# A recipe for scalable attention-based ML potentials: unlocking long-range accuracy with all-to-all node attention

**Eric Qu** [1] [*]  **Brandon M. Wood** [2]  **Aditi S. Krishnapriyan** [1] [3]  **Zachary W. Ulissi** [2]

## Abstract

Machine-learning interatomic potentials (MLIPs) have advanced rapidly, with many top models relying on strong physics-based inductive biases. However, as models scale to larger systems like biomolecules and electrolytes, they struggle to accurately capture long-range (LR) interactions, leading current approaches to rely on explicit physics-based terms or components. In this work, we propose AllScAIP, a straightforward, attention-based, and energy-conserving MLIP model that scales to O(100 million) training samples. It addresses the long-range challenge using an all-to-all node attention component that is data-driven. Extensive ablations reveal that in low-data/small-model regimes, inductive biases improve sample efficiency. However, as data and model size scale, these benefits diminish or even reverse, while all-to-all attention remains critical for capturing LR interactions. Our model achieves state-of-the-art energy/force accuracy on molecular systems, as well as a number of physics-based evaluations (OMol25), while being competitive on materials (OMat24) and catalysts (OC20). Furthermore, it enables stable, long-timescale MD simulations that accurately recover experimental observables, including density and heat of vaporization predictions.

## 1 Introduction

Machine-learning interatomic potentials (MLIPs) learn surrogate potential-energy surfaces from *ab initio* data to enable molecular dynamics at near–DFT fidelity and practical cost

[*]Work done at Meta. [1]Department of EECS, UC Berkeley [2]Fundamental AI Research (FAIR) at Meta [3]Applied Mathematics and Computational Research, LBNL. Correspondence to: Aditi S. Krishnapriyan <aditik1@berkeley.edu>, Zachary W. Ulissi <zulissi@meta.com>.

*Proceedings of the 43$^{rd}$ International Conference on Machine Learning*, Seoul, South Korea. PMLR 306, 2026. Copyright 2026 by the author(s).

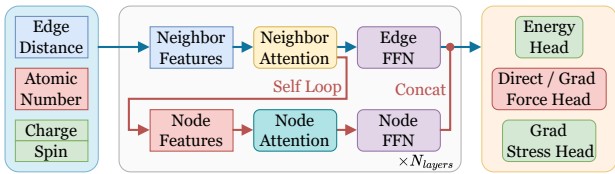

*Figure 1.* **AllScAIP model design.** The simple backbone design enables efficient scaling.

(Unke et al., 2021). The field has progressed from descriptor-based neural potentials (Behler & Parrinello, 2007) to invariant message-passing networks (Schütt et al., 2017; Gasteiger et al., 2021; 2022), SE(3)-equivariant architectures with high-order directional features (Batzner et al., 2022; Passaro & Zitnick, 2023; Fu et al., 2025), and more recently attention-based models (Orbital-Materials, 2024; Qu & Krishnapriyan, 2024; Mazitov et al., 2025). Across these approaches, designs differ chiefly in their inductive biases: symmetry (translation, permutation, and rotation invariance or equivariance), locality (cutoffs and neighbor stencils), smoothness/regularization, architectural constraints, and explicit physical structure (energy-conserving gradients). These choices reflect two perspectives: one emphasizes richer inductive biases to encode geometry and physics up front, which can gain sample efficiency on small datasets. The other treats some inductive biases as a scaffold that can be reduced as data and parameters grow, letting the model discover features while prioritizing training efficiency and stability.

As MLIPs have grown more capable, evaluation has shifted to larger and more complex systems. In these regimes, such as biomolecules and electrolytes systems, long-range (LR) interactions are important for describing subtle and important interactions. LR effects, such as electrostatics, induction/polarization, dispersion, electronic structure changes, couple atoms across long length scales. However, the most scalable MLIPs for very large datasets are message-passing networks built on local radius-graph constructions with distance cutoffs. A common approach to address these limitations is to add explicit physics-based inductive biases: attach polarization or charge-equilibration solvers (Ko et al., 2021); incorporate analytic or learned dispersion (e.g., many-body vdW) (Sauer et al., 2025); or introduce continuum/elastic corrections (Gong et al., 2025); predict per-atom charges (or multipoles) and evaluate Coulomb

terms via Ewald/PME or FMM (Loche et al., 2025; Kim et al., 2025). These strategies have been extensively validated on a number of targeted, small-scale datasets. However, as the field shifts towards models that can deliver high accuracy across large, heterogeneous datasets spanning many distinct systems, driven in part by the release of large-scale datasets (Levine et al., 2025), developing approaches that scale effectively to this setting remains an open challenge.

We hypothesize that several inductive biases are *learnable under scale*—notably rotational symmetry (Qu & Krishnapriyan, 2024; Mazitov et al., 2025), locality via radius graph structure (Kreiman et al., 2025), and long-range interactions—whereas others are harder to learn and may need to be encoded by the architecture or loss function (e.g. energy conservation via gradient-based forces). Guided by this view, we develop a conceptually straightforward, scalable, attention-based MLIP with two stages of operations (illustrated in Fig 1): *neighborhood self-attention* (based on the EScAIP model (Qu & Krishnapriyan, 2024)) operating on fixed local stencils to resolve local information, followed by an *all-to-all node self-attention* that mixes information globally, allowing signals to travel over the whole graph. Both stages use off-the-shelf multi-head self-attention operation CUDA kernels that have been highly engineered and optimized for popular AI/ML applications in computer vision and language (Lefaudeux et al., 2022). To test our hypothesis, we add two more ingredients to the recipe that provide features not directly captured by the architecture itself: *Legendre Angular Encoding (LAE)*, a compact, rotation-aware edge encoding that supplies high-order directional information to the neighborhood attention; and *Euclidean Rotary Position Encoding (ERoPE)* (based on Frank et al. (2024)), an isotropic distance-only encoding that injects distance information into the node attention. These serve as inductive biases that can be explicitly built into the model architecture. We ablate these components across model and data scales to test whether such geometric inductive biases are indeed learnable when model capacity and data size grow.

Across datasets and scales, the ablations support the "inductive biases learnable under scale" hypothesis. In the low-data/small-model regime, every ingredient is useful: adding LAE lowers force error by supplying directional/angle signals, the Euclidean RoPE improves energy with distance information, and the global node attention delivers the largest gains by enabling many-hop communication without deep stacks. As we increase both model capacity and dataset size, the picture shifts: the marginal benefit of the geometric encodings diminishes and sometimes reverses sign, indicating that angular and radial features can be learned end-to-end at scale. By contrast, the all-to-all node-attention stage remains the most durable source of long-range improvement.

In summary, our results support a prior-light recipe for scal-able MLIPs. In the regimes we evaluated, with sufficient data and parameters, several inductive biases, including rotational equivariance, high-order directional features, and even long-range interactions, are largely learnable. The resulting model: AllScAIP (**All**-to-all **Sc**alable **A**ttention **I**nteratomic **P**otential), attains state-of-the-art on both energy/force accuracy and relevant physics-based evaluations on the largest and most diverse molecules dataset to date, OMol25 (Levine et al., 2025). AllScAIP trained on OMol25 is able to perform stable, long-timescale MD simulations that accurately recover experimental observables, including density and heat of vaporization predictions. AllScAIP is also competitive on materials (OMat24) (Barroso-Luque et al., 2024) and catalysts systems (OC20) (Tran et al., 2023). Taken together, the results suggest that a data-driven path for MLIPs may be competitive with other approaches: prioritize scalable components, keep priors lightweight, and letting scale handle the rest.

## 2 Related Works

**Machine Learning Interatomic Potentials.** There have been significant advances in MLIP models, which are surrogate models that are trained to predict energies and forces from atomic numbers and positions. We group current approaches into two categories: (1) models that use equivariant node features, including NeuqIP (Batzner et al., 2022), MACE (Batatia et al., 2022), SCN (Zitnick et al., 2022), eSCN (Passaro & Zitnick, 2023), Equiformer (Liao & Smidt, 2022; Liao et al., 2024), eSEN (Fu et al., 2025), UMA (Wood et al., 2025); (2) models that use scalar node features, where they only enforce basic symmetries: SchNet (Schütt et al., 2017), DimNet (Gasteiger et al., 2020), GemNet (Gasteiger et al., 2021; 2022), EScAIP (Qu & Krishnapriyan, 2024), MindScAIP (Liu et al., 2026), OrbNet (Orbital-Materials, 2024), DPA4 (Zhang et al., 2025), MatRIS (Zhou et al.), PET-MAD (Mazitov et al., 2025).

**Long-range Interactions in MLIPs.** To enable the ability to model long range interaction in local GNN-based MLIPs, the main paradigm is to inject explicit long-range physics: models predict charges, multipoles, or surrogate charge densities and evaluate electrostatics with Ewald/PME/FMM, including PhysNet (Unke & Meuwly, 2019), 4G-HDNNP (Ko et al., 2021), AIMNet (Zubatyuk et al., 2019; Anstine et al., 2024), BAMBOO (Gong et al., 2024), LODE (Grisafi & Ceriotti, 2019), and DPLR (Zhang et al., 2022). Recent "latent" approaches learns a hidden variable and applying an Ewald-style long-range energy directly (Cheng, 2025).

## 3 Methods

### 3.1 Attention Operations

We treat attention as an off-the-shelf operation, taking advantage of the highly optimized CUDA kernels (Lefaudeux et al.,

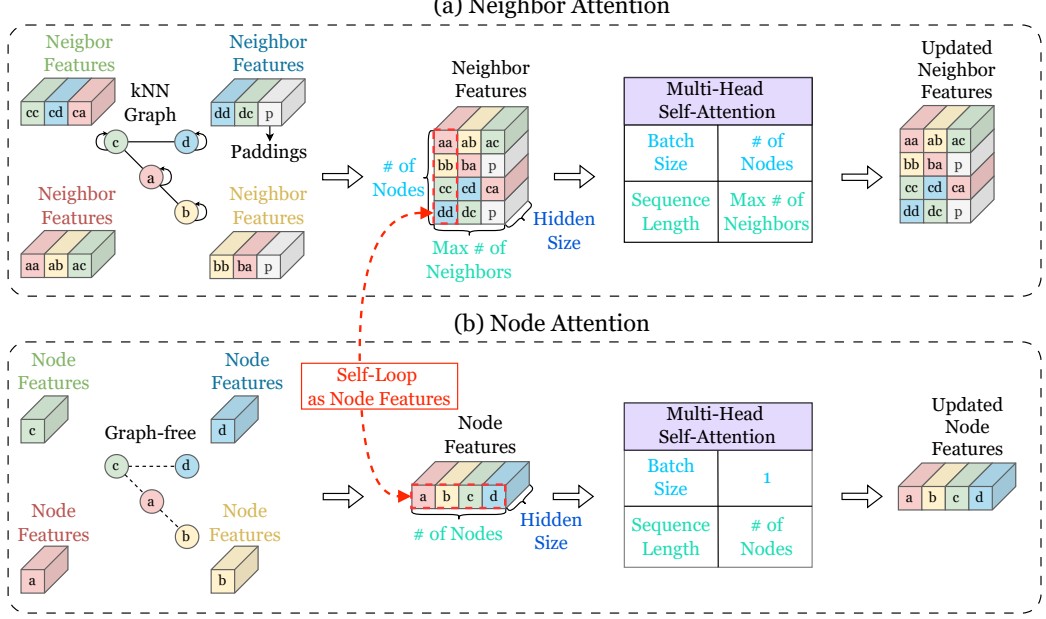

*Figure 2.* **Illustration of the attention operations used in AllScAIP.** (a) Neighbor attention. (b) Node attention.

2022). The only difference between the two stages is how we *pack tokens*: local (neighbor) attention operates on fixed neighbor list per node and scales as $\mathcal{O}(Nk)$, while all-to-all node attention mixes all nodes per graph with $\mathcal{O}(N^2)$ cost.

**Neighborhood Self-attention.** Following the EScAIP model (Qu & Krishnapriyan, 2024), each center atom gathers up to $k$ neighbors plus a self token, yielding a tensor of shape $(\#\text{nodes}) \times (k+1) \times d_{\text{model}}$ that we feed to standard multi-head self-attention (MHSA) (Fig 2 (a)). We run two directional passes over the same stencil: center→neighbors (out) and neighbors→center (in). A smooth distance-based envelope provides a padding/mask that softly fades far pairs.

**All-to-all Node Self-attention.** Local neighborhoods resolve fine geometry, but long-range interactions require global communication. We therefore apply MHSA over the node stream by packing the nodes to $1 \times (\#\text{nodes}) \times d_{\text{model}}$ and using the same operator (Fig 2 (b)). In practice, this stage complements the local passes: neighborhood attention handles fine, anisotropic interactions; node attention enables many-hop coupling in a single step.

### 3.2 Geometric Encodings

In both neighborhood and node attention, each entry in the attention matrix corresponds to a geometric pair: an edge–edge interaction in the neighborhood case and a node–node interaction in the global case. We exploit this by injecting geometric encoding into the attention logits. Legendre Angular Encoding provides directional (angles), and Euclidean RoPE provides radial (distances) signals that are not explicitly present in the vanilla Q/K/V streams; in principle, a suffi-

ciently large model could learn them from coordinates. It's an important choice whether to include them as input (inductive biases) or let the model directly learn from data/scaling.

**Legendre Angular Encoding (LAE).** This encoding is used to inject angular and high-order directional features to the neighborhood attention logits. Each directed edge $i \rightarrow j$ is assigned a compact angular code built from real spherical harmonics evaluated on the unit direction $\hat{\mathbf{r}}_{ij}$. For degrees $\ell = 0, ..., L$ we compute $\mathbf{Y}_\ell(\hat{\mathbf{r}}_{ij}) \in \mathbb{R}^{2\ell+1}$, repeat each degree block $r_\ell$ times, and concatenate across $\ell$ to obtain $\gamma_{ij} \in \mathbb{R}^{d_{h_r}}$ with $d_{h_r} = \sum_\ell r_\ell(2\ell+1)$. We choose $\{r_\ell\}$ such that the sum matches the per-head dimension: $\sum_{\ell=0}^{L} r_\ell = d_h$. For head $h$, the queries/keys are repeated $2\ell+1$ times accordingly and modulated elementwise by $\gamma_{ij}$,

$$\tilde{q}_{i,h} = \bar{q}_{i,h} \odot \gamma_{ij}, \quad \tilde{k}_{j,h} = \bar{k}_{j,h} \odot \gamma_{ij},$$

and the usual scaled dot-product is applied, $a_{ij,h} = (\tilde{q}_{i,h}^\top \tilde{k}_{j,h})/\sqrt{d_h}$, followed by standard attention. By the spherical-harmonic addition theorem, inner products within each degree correspond (up to a constant) to $P_\ell(\cos\theta)$, so LAE supplies multi-order angular structure with linear cost. LAE is rotation-aware, parameter-light, and integrates with off-the-shelf attention kernels (Fig. 3 (a)).

**Euclidean Rotary Position Encoding (ERoPE).** To inject distance information in the node attention attention logits, we add an isotropic radial bias. Following Frank et al. (2024), for each pair $(i,j)$ with separation $r_{ij}$, we build a purely radial encoding by averaging plane-wave phases over all orientations on the unit sphere, which yields the isotropic kernel $K_\omega(r) = \text{sinc}(\omega r) = \sin(\omega r)/(\omega r)$.

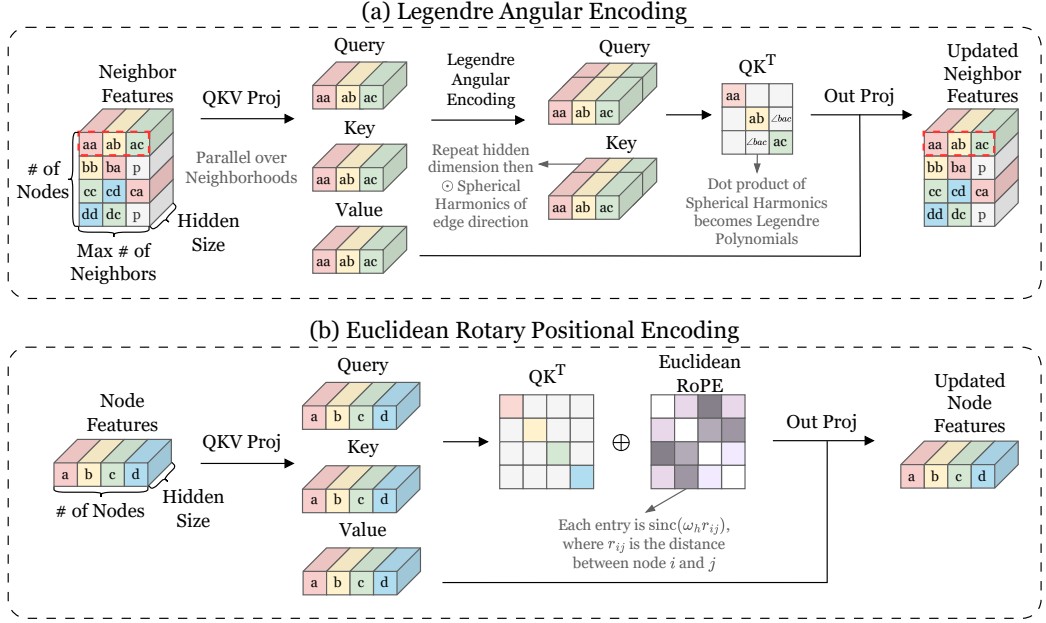

*Figure 3.* **Illustration of the geometric encoding used in AllScAIP.** (a) Legendre Angular Encoding (LAE) (b) Euclidean rotary position encoding.

We choose a bank of $M$ log-spaced frequencies $\{\omega_k\}_{k=1}^M \subset [\omega_{\min}, \omega_{\max}]$ (shared across heads), evaluate $\mathbf{s}_{ij} = [\operatorname{sinc}(\omega_1 r_{ij}), \ldots, \operatorname{sinc}(\omega_M r_{ij})]^\top$. Each head $h$ mixes them with a learned weight vector $\mathbf{w}_h \in \mathbb{R}^M$ to produce an additive logit bias $b_{ij,h} = \mathbf{w}_h^\top \mathbf{s}_{ij}$, and the final logit is

$$a_{ij,h} = \frac{q_{i,h}^\top k_{j,h}}{\sqrt{d_h}} + b_{ij,h}.$$

Because $b_{ij,h}$ depends only on $r_{ij}$, the bias is translation- and rotation-invariant by construction. The small frequency bank adds only $\mathcal{O}(M)$ work per pair, can be precomputed per batch, and provides a smooth, spectrally rich radial prior without altering the softmax mechanism (Fig. 3 (b)).

### 3.3 AllScAIP Model

Fig. 1 shows the whole architecture of AllScAIP. The input are initialized from atom numbers, edge distances RBF, and optional charge/spin. Each block first applies neighborhood attention on the neighbor stream (with optionally added LAE encoding). This local stage is followed by RMSNorm, residual path, and an edge FFN. The block then takes self-loop neighbor features as node features, and performs all-to-all node attention on the node stream (with optional Euclidean RoPE), followed by RMSNorm, residual path, and a node FFN. The updated node features are concatenated with the neighbor features and carried to the next layer. After $N_{\text{layers}}$ blocks, the node stream feeds an energy head; forces are supervised either directly or via energy gradients (energy-conserving option), and a gradient-based stress head is included when required.

### 3.4 Inductive Biases

We distinguish **hard** priors we encode by architecture, **soft** priors we can optionally supply as features, and **learnable** physical structure that we leave to scale. The goal is a prior-light model that keeps only what is hard to learn with high precision (locality, conservation, permutation/translation symmetries) while letting large models and datasets learn the rest. We present the inductive biases we have considered in Tab. 1. In short, we enforce core symmetries (translation, permutation), locality, energy conservation, and extensivity; we optionally inject light geometric features; and we leave rotation and long-range interaction to be learned with scale. The energy conservation formulation follows the differentiable kNN algorithm in Liu et al. (2026). We validate these properties in § 5.2.

## 4 Ablations

**Ablations on Model Components.** We begin by isolating the contribution of each architectural piece at a fixed capacity and data scale. We use the medium configuration (AllScAIP-md, about 85M parameters) trained on OMol25 4M split (Levine et al., 2025) for 80 epochs with direct force supervision. We toggle neighborhood self-attention (NeiAtt; always on), Legendre Angular Encoding (LAE), the all-to-all node attention (NodeAtt), and the Euclidean radial bias (ERoPE). We report Energy/Atom MAE (mEV) and Force MAE (meV/Å) on the validation set for total and four splits: Biomolecules, Electrolytes, Metal Complexes, and Neutral Organics (Tab. 2, top 4M section).

*Table 1.* **Inductive Biases in AllScAIP Model**

| Inductive Biases | Status | How satisfied / where |
|---|---|---|
| Translation invariance | **Enforced** | Use only relative geometry (distances/directions). |
| Permutation equivariance | **Enforced** | Attention over sets with shared weights. |
| Extensivity (PBC) | **Enforced** | Sum aggregation for energy with PBC minimal-image distances. |
| Extensivity (non-PBC) | *Learnable* | Not hard-coded for the all-to-all node attention (see App. A.5). |
| Locality prior | **Enforced** | kNN graph with smooth cutoff in neighborhood attention. |
| Energy conservation | **Enforced** | Grad-based forces: $F = -\mathrm{d}E/\mathrm{d}x$; differentiable kNN graph. |
| Rotational equivariance | *Learnable* | Not hard-coded; can be learned under scale. |
| Long-range effects | *Learnable* | All-to-all node attention provides an infinite receptive field. |
| Radial/distance prior | *Optional (soft)* | **Euclidean RoPE**: distance-based encoding in node attention. |
| High-order directional features | *Optional (soft)* | **LAE**: compact spherical harmonic encoding modulates Q/K in neighborhood attention (angles/directions). |

*Table 2.* **Component ablations under data and model size scaling**. We report Energy / Atom MAE (meV) and Force MAE (meV/Å), lower is better. Each size's **with encoding** (Nei Att + LAE + Node Att + ERoPE) configuration is the reference; other rows are colorized relative to that reference (Green = better, Red = worse). The Throughput column reports inference speed (ns/day on one H200 with 1000 atoms). We report results for the OMol25 4M split (80 epochs) and the full 102M (10 epochs)[1]

| | | **Ablations** | | | | **OMol 4M (80 Epochs)** | | | | | | | | | |
|---|---|---|---|---|---|---|---|---|---|---|---|---|---|---|---|
| | | | | | | Biomol. | | Elytes. | | Metal Cplx. | | Neutral Org. | | Total | |
| Size | Thrpt | NeiAtt | LAE | NodeAtt | ERoPE | E↓ | F↓ | E↓ | F↓ | E↓ | F↓ | E↓ | F↓ | E↓ | F↓ |
| 35M | 2.279 | ✓ | ✓ | ✓ | ✓ | 0.38 | 5.57 | 1.27 | 8.93 | 2.85 | 33.52 | 1.89 | 13.52 | 1.46 | 9.35 |
| | 7.623 | ✓ | | ✓ | | 0.5 | 6.32 | 1.51 | 9.81 | 2.96 | 34.56 | 2.23 | 14.51 | 1.75 | 10.21 |
| 85M | 1.124 | ✓ | ✓ | ✓ | ✓ | 0.23 | 3.51 | 1.05 | 6.38 | 2.46 | 27.83 | 1.21 | 8.93 | 1.16 | 6.88 |
| | 3.333 | ✓ | | ✓ | ✓ | 0.29 | 4.27 | 1.17 | 7.22 | 2.67 | 29.79 | 1.53 | 10.53 | 1.22 | 7.6 |
| | 1.392 | ✓ | ✓ | ✓ | | 0.29 | 3.78 | 1.16 | 6.67 | 2.58 | 28.26 | 1.36 | 9.03 | 1.3 | 7.1 |
| | 4.014 | ✓ | | ✓ | | 0.32 | 4.24 | 1.17 | 7.16 | 2.64 | 29.19 | 1.57 | 10.12 | 1.25 | 7.55 |
| | 1.281 | ✓ | ✓ | | | 0.52 | 4.58 | 1.39 | 7.42 | 2.81 | 29.86 | 1.43 | 10.01 | 1.56 | 7.93 |
| | 4.327 | ✓ | | | | 0.55 | 4.96 | 1.45 | 7.92 | 2.85 | 30.78 | 1.57 | 10.77 | 1.72 | 8.48 |
| Size | Thrpt | NeiAtt | LAE | NodeAtt | ERoPE | **OMol 102M (10 Epochs)** | | | | | | | | | |
| 35M | 2.279 | ✓ | ✓ | ✓ | ✓ | 0.21 | 3.85 | 0.75 | 6.01 | 2.09 | 26.02 | 1.01 | 8.54 | 0.85 | 6.61 |
| | 7.623 | ✓ | | ✓ | | 0.29 | 4.47 | 0.81 | 6.73 | 2.12 | 26.86 | 1.08 | 9.17 | 0.98 | 7.22 |
| 85M | 1.124 | ✓ | ✓ | ✓ | ✓ | 0.15 | 2.81 | 0.53 | 4.59 | 1.83 | 21.93 | 0.73 | 6.03 | 0.67 | 5.1 |
| | 3.333 | ✓ | | ✓ | ✓ | 0.2 | 3.16 | 0.55 | 4.99 | 1.83 | 22.48 | 0.79 | 6.47 | 0.73 | 5.51 |
| | 1.392 | ✓ | ✓ | ✓ | | 0.2 | 2.93 | 0.58 | 4.76 | 1.8 | 21.59 | 0.76 | 5.94 | 0.72 | 5.13 |
| | 4.014 | ✓ | | ✓ | | 0.15 | 2.91 | 0.52 | 4.7 | 1.83 | 22.31 | 0.72 | 6.29 | 0.64 | 5.23 |
| | 4.327 | ✓ | | | | 0.4 | 3.83 | 0.79 | 5.57 | 2.01 | 23.91 | 0.85 | 7.05 | 0.96 | 6.15 |

Removing LAE consistently worsens the force performance across all splits, while its effect on energies is smaller. This matches the role of LAE as a directional/angle feature: adding orientation information improves how the model fit to vector targets. In contrast, turning off ERoPE primarily harms energies and has a milder impact on forces, consistent with ERoPE's distance-only, isotropic bias acting as a radial prior on scalar predictions. Finally, ablating the all-to-all node-attention degrades both energy and force, with especially pronounced gaps in biomolecules. The full configuration achieves the best results across all splits, confirming that the module improves sample efficiency in the low data regime.

**Ablations on Data and Model Size Scaling.** To quantify how inductive bias interacts with scale, we vary two axes:

data (OMol 4M → OMol 102M) at fixed capacity, and model size (35M → 85M) at fixed data.

Under **data scaling** (4M → 102M) at fixed capacity, energy and force errors decrease across domains. In many cases, removing the fixed geometric encodings matches or slightly improves upon the full model at the same scale, suggesting that angular/radial signals can be learned directly from larger data. The all-to-all node attention remains consistently useful, indicating that a global mixing stage is a durable mechanism for long-range interactions.

Under **model-size scaling** (35M → 85M) at fixed data, absolute errors also drop with capacity. At 4M, geometric encodings still help, but their benefit diminishes as model size grows; at 102M, medium models show parity or slight disadvantages for fixed encodings, while retaining a

*Table 3.* **Symmetry and Conservation Checks.** Extensivity errors for periodic (PBC) and non-periodic (non-PBC) systems, (lower is better), rotational equivariance via force cosine similarity under random rotations (higher is better), and NVE molecular dynamic simulation energy drift on MD22 large molecules (lower is better).

| Model | Training Set | PBC Energy $\Delta$ [meV] $\downarrow$ | non-PBC Energy $\Delta$ [meV] $\downarrow$ | Rand. Rotation Cos. Sim. $\uparrow$ | NVE MD Energy Drift [meV / atom / ps] $\downarrow$ |
|---|---|---|---|---|---|
| UMA-M-1p1 | UMA-459M | $2.41 \times 10^{-2}$ | $3.83 \times 10^{-3}$ | 0.9999 | $1.5 \times 10^{-3}$ |
| AllScAIP-md | NA (Random Init) | $3.62 \times 10^{-2}$ | $2.49 \times 10^{-3}$ | 0.6827 | - |
| AllScAIP-md | OMol-102M | $4.18 \times 10^{-2}$ | $4.73 \times 10^{-3}$ | 0.9999 | $4.3 \times 10^{-3}$ |

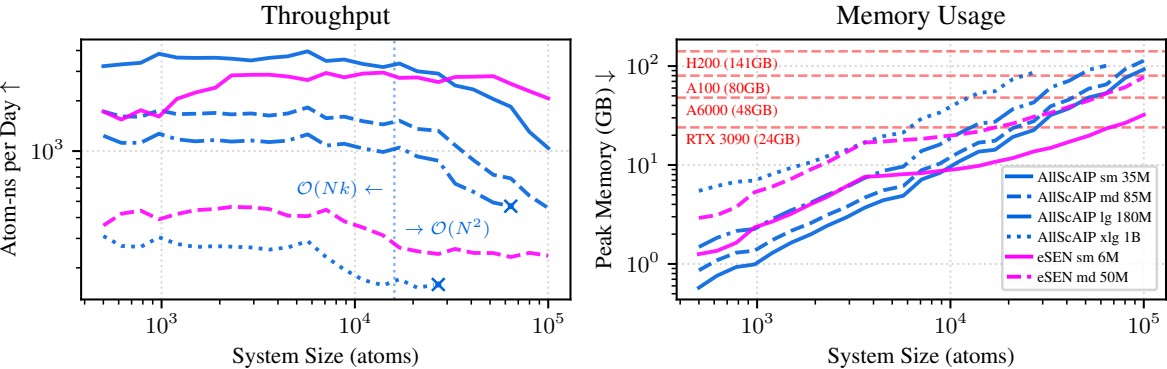

*Figure 4.* **Throughput and Memory vs. System size.** Left: Atom-time (number of atoms $\times$ ns/day, higher is better) vs. system size. The result is measured on a single H200 141G with graph generation off. We report four model sizes (35M/85M/180M/1B) of AllScAIP, and eSEN baselines. The dotted vertical lines indicate approximately when the $\mathcal{O}(N^2)$ of the node attention dominates over the $\mathcal{O}(Nk)$ of the neighborhood attention, where $k$ is the max number of neighbors. Right: vram usage vs. system size. Dotted horizontal lines indicate the vram size for common GPUs.

measurable advantage from the all-to-all attention. In short: as data and parameters scale, the marginal value of fixed geometric encodings decreases, whereas the architectural affordance of global mixing continues to pay off.

## 5 Results

### 5.1 Inference Efficiency and System–size Scaling

We test the efficiency and memory scaling of the AllScAIP model. Fig. 4 plots atom–ns/day versus system size that exposes a slope change. In the small $N$ regime, speed is dominated by neighborhood attention ($\mathcal{O}(Nk)$), atom–ns/day is nearly flat or linear in $N$; once the all-to-all node attention dominates ($\mathcal{O}(N^2)$), it falls roughly as $1/N$ (slope $-1$ in log-log). The dashed vertical lines mark this empirical transition. Overall, the curves show predictable scaling and our models are still efficient compared with baselines. Our method targets around $10^3 - 10^5$ atoms, where much interesting science lives (biomolecules, electrolytes, soft matter, mesoscale crystallites) and GPUs remain efficient. In addition, we also provide the raw throughput vs. system size at Fig. 9, and per-component breakdowns at Fig. 8.

### 5.2 Symmetry & Energy Conservation Checks

We verify that AllScAIP satisfies key inductive-bias properties and compare to a strong baseline trained on various chemical systems—UMA (Wood et al., 2025) (Tab. 3).

**Extensivity.** We test extensivity in both periodic and non-periodic settings:

(i) **PBC supercell doubling:** given a periodic structure, we build a $2\times$ supercell by translating the cell by one lattice vector and recomputing the energy; extensivity requires $E_{2\times} \approx 2E_{1\times}$. We report the $E_\Delta = |E_{2\times} - 2E_{1\times}|$.

(ii) **Vacuum duplication:** for a non-periodic system, we duplicate the configuration and translate the copy by a large offset $R$, then evaluate $E(R)$; as $R \to \infty$ the fragments are non-interacting and $E(R) \to 2E_{\text{single}}$. In practice, we use $R = 1000\text{Å}$ and report the deviation $E_\Delta = |E(R) - 2E_{\text{single}}|$.

In the PBC setting, extensivity is enforced by construction via sum aggregation with minimal-image distances. In the non-PBC setting, the all-to-all node attention does not strictly guarantee additivity; instead, we rely on ReZero-initialized residuals (Bachlechner et al., 2021) for the node-attention block, so its contribution starts at zero and is shaped by training. Empirically, the trained model attains near-perfect additivity in both settings, indicating that the long-range coupling is learned without introducing unphysical residual interactions at large separation. We provide additional evidence in App. A.5.

**Rotational Equivariance.** We sample 1000 OMol test structures, draw a random rotation $R \in \text{SO}(3)$, and evaluate forces twice: $\mathbf{F}^{(1)}$ on the original coordinates and $\mathbf{F}^{(2)}$

*Table 4.* **OMol25 validation results.** Energy / Atom MAE (meV) and Force MAE (meV/Å) across Biomolecules, Electrolytes, Metal Complexes, and Neutral Organics. AllScAIP variants achieve the lowest overall energy error on both splits, with competitive force errors.

| Dataset | Model | Biomol. E↓ | Biomol. F↓ | Elytes. E↓ | Elytes. F↓ | Metal Cplx. E↓ | Metal Cplx. F↓ | Neutral Org. E↓ | Neutral Org. F↓ | Total E↓ | Total F↓ |
|---|---|---|---|---|---|---|---|---|---|---|---|
| | eSEN-sm-d. | 0.67 | 6.30 | 1.24 | 9.41 | 2.53 | 33.08 | 1.23 | 13.84 | 1.49 | 9.92 |
| | eSEN-sm-cons. | 0.59 | 4.61 | 1.01 | 8.08 | 2.30 | 28.86 | 0.84 | 11.11 | 1.27 | 8.25 |
| | eSEN-md-d. | 0.34 | **2.61** | 0.69 | **4.40** | **1.73** | **19.99** | **0.59** | **5.63** | 0.84 | **4.76** |
| All | GemNet-OC | **0.15** | 3.88 | 0.56 | 5.98 | 1.83 | 25.12 | 0.86 | 10.38 | 0.66 | 6.52 |
| | AllScAIP-sm-ft-cons. | 0.22 | 3.84 | 0.75 | 6.01 | 2.09 | 26.02 | 1.00 | 8.55 | 0.85 | 6.61 |
| | AllScAIP-md-d. | **0.15** | 2.91 | **0.52** | 4.70 | 1.83 | 22.31 | 0.72 | 6.29 | **0.64** | 5.24 |
| | AllScAIP-md-cons. | 0.22 | 3.23 | 0.53 | 5.29 | 1.86 | 22.40 | **0.59** | 7.55 | 0.67 | 5.78 |
| | eSEN-sm-d. | 0.88 | 8.12 | 1.93 | 12.64 | 3.37 | 40.44 | 2.16 | 20.17 | 2.19 | 13.01 |
| | eSEN-sm-cons. | 0.86 | 6.17 | 1.61 | 11.16 | 2.72 | 35.33 | 1.50 | 16.92 | 1.89 | 11.10 |
| 4M | eSEN-md-d. | 0.47 | 3.38 | 1.18 | **6.51** | 2.53 | **27.31** | 1.21 | **9.26** | 1.32 | **6.78** |
| | GemNet-OC | 0.25 | 5.20 | 1.04 | 8.42 | 2.66 | 32.76 | 1.64 | 15.59 | 1.13 | 8.98 |
| | AllScAIP-sm-ft-cons. | 0.27 | 4.06 | 0.91 | 7.81 | 2.26 | 32.10 | 1.11 | 12.73 | 1.04 | 8.19 |
| | AllScAIP-md-ft-cons. | **0.20** | **3.01** | 0.75 | 6.79 | **2.06** | 29.41 | 0.78 | 9.45 | **0.90** | 7.67 |

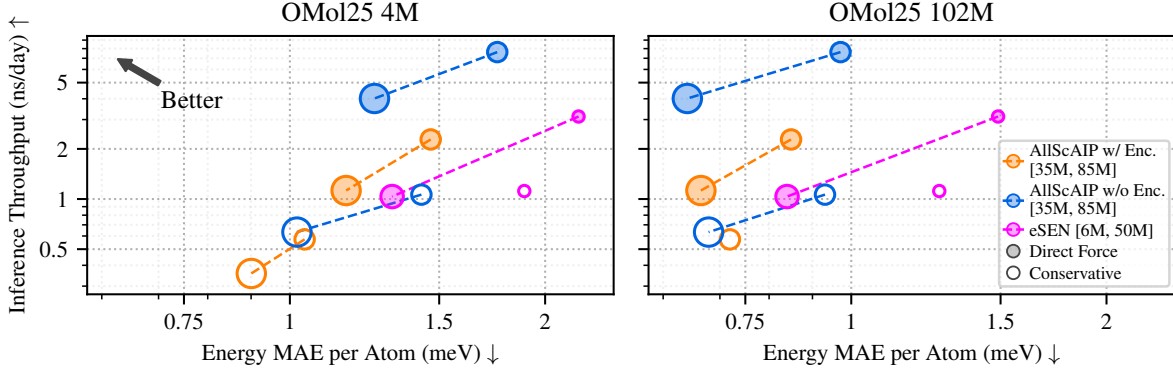

*Figure 5.* **OMol25 energy error vs. efficiency throughput (inference).** Energy / Atom MAE (meV, ↓) vs. throughput (ns/day, ↑) for 35M/85M models with/without encodings; compared with eSEN (Fu et al., 2025) baselines. Conservative models are labeled with a hollow marker. Left: Models trained on OMol25 4M (80 epochs). Right: OMol25 102M (12 epochs). Our models trace the Pareto front at 4M; at 102M the gap between with/without encodings shrinks or flips, indicating these inductive bias may be unnecessary at scale.

on the rotated atom position $R\mathbf{X}$. We then rotate the first prediction, $R\mathbf{F}^{(1)}$, and compute the similarity with $\mathbf{F}^{(2)}$, averaging over atoms and structures. AllScAIP trained on OMol-102M achieves a cosine similarity of 0.9999, whereas a randomly initialized model yields a much lower value. This is consistent with concurrent findings on non-equivariant architectures (Shen et al., 2025), indicating that approximate rotational equivariance *emerges in practice* from training at scale in the regime we tested.

**Energy Conservation.** We follow the differentiable kNN algorithm in Liu et al. (2026) to achieve energy conservation. Following Fu et al. (2025), we run NVE MD simulations on the seven large MD22 molecules (Chmiela et al., 2023) for 100 ps and report the energy drift (eV/atom/ps). AllScAIP shows small drift, comparable in scale to the UMA model.

### 5.3 Open Molecules (OMol25)

**Settings.** We train AllScAIP on the OMol25 (Levine et al., 2025) dataset under the following settings:

a. **4M split:** (i) direct force training for 80 epochs (**–d.**); (ii) a conservative fine-tune where we train direct force for 50 epochs, then swap to a gradient-based energy / force head and continue for 30 epochs (**–ft-cons.**).
b. **Full 102M:** (i) direct force for 10 epochs (**–d.**); (ii) direct force for 10 epochs + 2-epoch conservative fine-tune (**–ft-cons.**); (iii) fully conservative training from scratch (**–cons.**).

All models are trained on V100 32G with full fp32.

**Energy and Force Accuracy.** Across OMol25, our models sit on the accuracy-speed Pareto (Fig. 5) and deliver state-of-the-art energy error with competitive force error (Table 4). On the 4M split, the conservative fine-tuned model attains the best overall energy; the direct-force variant trades a small energy error increase for better forces. On the full dataset, the direct force model achieves the best overall energy accuracy. The same trends hold across all four splits, with especially strong gains on **biomolecules**, where long-range interactions is most critical.

*Table 5.* **Evaluation results across OMol25 test evaluations.** Results are reported in energy MAE error [meV] (lower is better). Models are sorted by the average ranking of individual categories, with a lower average ranking indicating better overall performance.

| Model | Energy Cons. | Training Set | Avg. Rank | Ligand pocket Ixn Energy ↓ | Ligand strain Strain Energy ↓ | Conformers ΔEnergy ↓ | Protonation ΔEnergy ↓ | Dist. scaling ΔEnergy ↓ | IE/EA ΔEnergy ↓ | Spin gap ΔEnergy ↓ |
|---|---|---|---|---|---|---|---|---|---|---|
| AllScAIP-md-d. | | OMol-102M | 3.43 | 29.496 | 4.149 | 4.985 | 16.104 | 10.082 | 143.035 | 350.825 |
| AllScAIP-md-cons. | ✓ | OMol-102M | 4.14 | 50.532 | 4.103 | 3.874 | 17.092 | 36.145 | 148.976 | 317.400 |
| eSEN-md-d. | | OMol-102M | 4.14 | 64.246 | 3.113 | 3.566 | 14.988 | 109.767 | 145.776 | 303.883 |
| UMA-M-1p1 | ✓ | UMA-459M | 4.86 | 76.778 | 2.962 | 2.532 | 18.457 | 138.094 | 136.386 | 335.058 |
| UMA-S-1p2 | ✓ | UMA-520M | 5.00 | 66.604 | 4.365 | 3.738 | 18.914 | 32.174 | 176.939 | 246.565 |
| AllScAIP-sm-ft-cons. | ✓ | OMol-102M | 6.29 | 58.450 | 4.770 | 4.106 | 27.515 | 15.473 | 168.858 | 341.708 |
| AllScAIP-md-ft-cons. | ✓ | OMol-4M | 6.43 | 46.727 | 4.839 | 5.172 | 19.399 | 16.473 | 169.104 | 367.133 |
| AllScAIP-sm-ft-cons. | ✓ | OMol-4M | 8.71 | 78.260 | 4.559 | 5.343 | 24.614 | 23.492 | 198.899 | 401.358 |
| GemNet-OC-r12 | | OMol-102M | 9.71 | 19.373 | 8.381 | 12.049 | 31.645 | 77.074 | 177.457 | 373.831 |
| eSEN-sm-cons. | ✓ | OMol-102M | 10.57 | 147.261 | 4.656 | 4.487 | 21.770 | 196.964 | 222.827 | 391.469 |
| GemNet-OC-r6 | | OMol-102M | 10.86 | 47.022 | 9.554 | 11.479 | 36.925 | 80.683 | 199.721 | 370.975 |
| UMA-S-1p1 | ✓ | UMA-459M | 11.14 | 127.723 | 4.856 | 5.170 | 27.969 | 194.688 | 206.872 | 369.181 |
| eSEN-md-d. | | OMol-4M | 11.14 | 98.337 | 4.508 | 5.369 | 24.975 | 111.063 | 240.212 | 430.753 |
| GemNet-OC-r12 | | OMol-4M | 12.86 | 36.221 | 12.489 | 19.797 | 50.773 | 99.995 | 237.682 | 490.463 |
| mace-omol-L-0 | ✓ | OMol-4M | 14.29 | 297.404 | 7.954 | 6.782 | 25.107 | 244.834 | 338.795 | 438.891 |
| eSEN-sm-cons. | ✓ | OMol-4M | 14.43 | 243.883 | 5.850 | 6.811 | 35.754 | 232.839 | 293.120 | 478.242 |
| GemNet-OC-r6 | | OMol-4M | 15.00 | 96.043 | 13.270 | 23.355 | 52.160 | 83.296 | 329.993 | 559.080 |

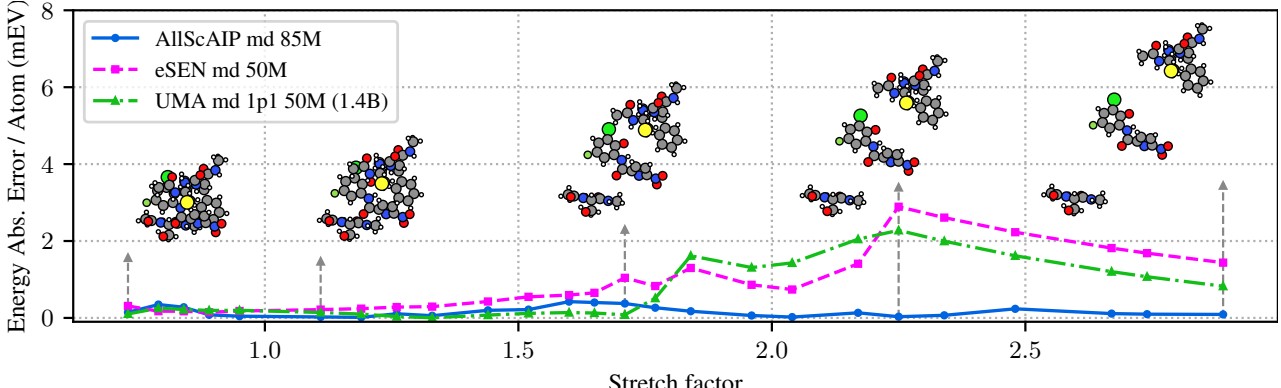

*Figure 6.* **OMol25 distance-scaling evaluation.** A group of molecules are uniformly compressed and stretched by a scalar "stretch factor" to probe the long-range ability (x-axis; < 1: compressed, > 1: stretched) and report energy absolute error per atom (meV; ↓). AllScAIP-md-cons. stays flat and low across the full range, while eSEN and UMA md-1p1 degrade sharply under stretching, indicating poor long-range capacities. Insets show example geometries at selected factors labeled with dotted lines.

**OMol25 Test Evaluations.** We benchmark on the full OMol25 evaluation suite (Tab. 5), sorting models by their average ranking across seven categories. AllScAIP-md-d. trained on the full 102M split tops by average rank. In particular, our model excels on the distance scaling (LR) test (about 90% reduction compared with the second best model): when molecules are uniformly compressed/stretched, energy error for AllScAIP remains low and flat, while eSEN and UMA degrade markedly under large stretches (Fig. 6). These results indicate that the proposed architecture transfers beyond the training distribution, providing robust long-range behavior while maintaining competitive accuracy on the remaining benchmarks (ligand interaction/strain, conformers, protonation, IE/EA, and spin gaps). Details of the evaluations can be found in Levine et al. (2025).

**MD Simulation and Comparison to Experiment.** To test how well AllScAIP performs when running molecular dynamics and computing macroscopic observables from the simulation, we ran out-of-the-box NPT simulations with the pre-trained model (no MD-specific fine-tuning). Stresses come from taking the energy gradient from the

displacements (not trained during training). We used a 1 fs timestep and ran NPT MD simulation for 300 ps, then computed densities and enthalpy of vaporization from the final 150 ps (after equilibration).

Following Kim et al. (2025), we ran NPT MD on 39 molecular liquid systems at 298K, and NVT MD on the corresponding isolated molecules to extract the potential energies for the enthalpy of vaporization (Fig. 7). We use AllScAIP trained on OMol25 to compare with eSEN, and use the model trained on SPICE (Appendix C) to compare with MACELES. AllScAIP achieves better MAE and $R^2$ compared with MACELES. The eSEN points show a systematic over-prediction (compression), whereas AllScAIP removes this bias.

Overall, the MD results indicate that the data-driven all-to-all node attention yields realistic bulk behavior without explicit long-range terms, consistent with our long-range ablations and distance-scaling tests. Additional results on MD simulations with the MD22 (Chmiela et al., 2023) datasets can be found in the Appendix E.

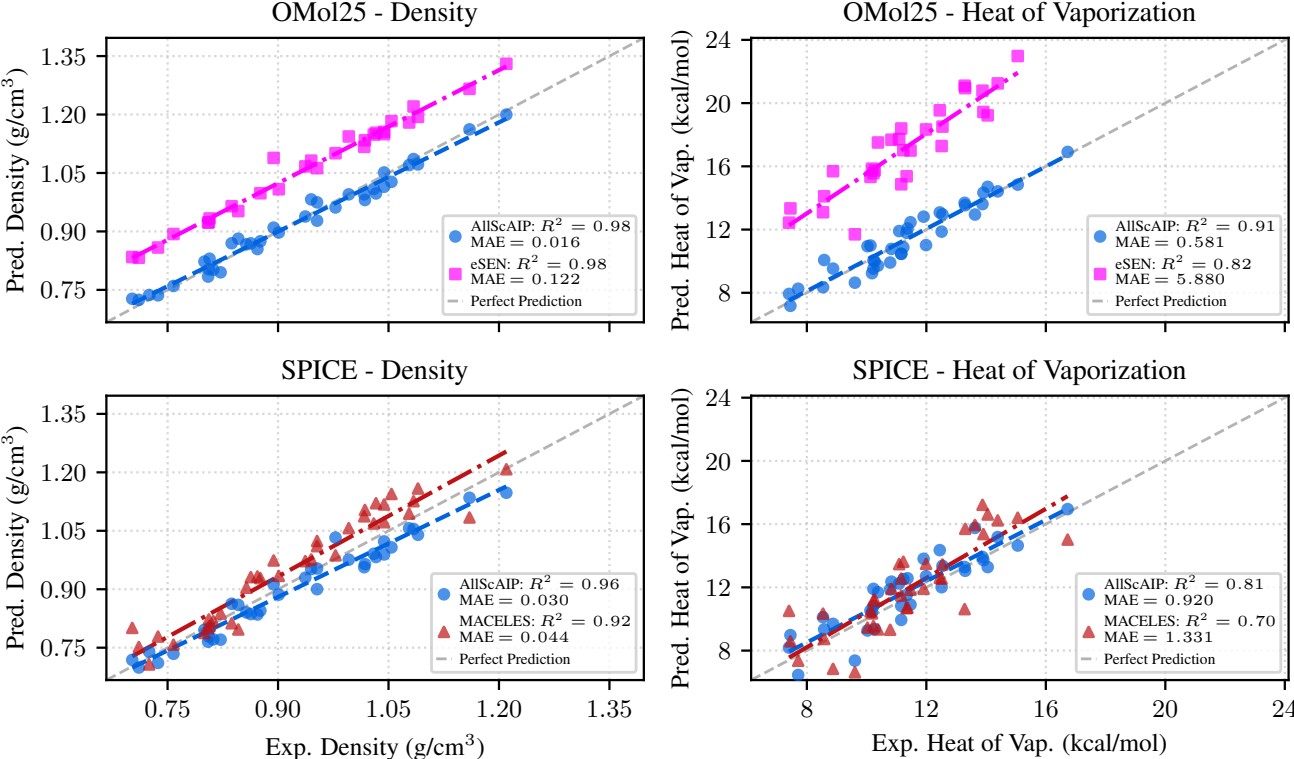

*Figure 7.* **MD Simulations vs. Experimental Density and Enthalpy of Vaporization.** NPT MD on 39 different molecular liquids and NVT MD on isolated molecules for 300 ps. Left: predicted vs. experimental densities. Right: predicted vs. experimental enthalpy of vaporization. Top: models trained on OMol25. Bottom: models trained on SPICE (note that SPICE structures, computed at a different DFT level of theory, is also a subset of OMol25). AllScAIP attains lower MAE and higher $R^2$ than MACELES and eSEN on both properties.

## 5.4 Open Materials and Open Catalyst

AllScAIP achieves competitive performance on Open Materials (OMat24) (Barroso-Luque et al., 2024) and Open Catalyst (OC20) (Chanussot et al., 2021). These are large-scale materials datasets that are very different in nature from OMol25. See Appendix B for details.

## 6 Discussions

**Scaling and the role of inductive bias.** Our ablations suggest a simple guideline: *scale first, bias second*. Directional/radial encodings (LAE, Euclidean RoPE) improve data efficiency in low-data/small-model regimes, but their marginal benefits diminishes as both data and parameters scale. In contrast, the architectural capacity for long-range coupling provided by the all-to-all node attention remains beneficial at every scale we tested. We also note that there are other works that challenge the graph as an inductive bias (Kreiman et al., 2025). Practically, this argues for prioritizing scalable, expressive components, because data and compute will only increase; such components benefit monotonically with scale, whereas fixed inductive features may cap flexibility.

**Data-driven vs. constrained paths for MLIPs.** A parallel line of work encodes direction with SE(3)–equivariant spherical-harmonic (SH) irreps. This route provides a rich prior directional basis, but in practice it is *band-limited*: after truncating at degree $L$, very sharp or dataset-specific angular features (e.g. narrow hydrogen-bond cones, $\pi$-stacking orientations, etc.) require larger $L$. The representation is powerful yet rigid: capacity is tied to the chosen SH order and coupling rules. Our path is different: we keep the backbone scalar and data-driven. As scale grows, it can allocate capacity where needed without being constrained by an SH decomposition, preserving hardware efficiency and flexibility.

**Limitations and opportunities.** The efficiency study shows a clear transition from local $\mathcal{O}(Nk)$ to $\mathcal{O}(N^2)$ complexity. We view $\mathcal{O}(N^2)$ as a manageable trade-off for long-range accuracy, especially given the engineering playbook that enabled very long context in LLMs: tiled/streaming attention (FlashAttention, ring attention) and activation checkpointing, which we already leverage through off-the-shelf attention kernels (Lefaudeux et al., 2022) and which keep peak memory well below the naive $\mathcal{O}(N^2)$ bound. Beyond systems work, several modeling routes can further delay the $\mathcal{O}(N^2)$ regime: hierarchical node pools with cross-pool attention; linear-time attention; and mixtures-of-experts that route only a fraction of nodes to global mixing.

## Acknowledgements

This work was partially supported by Scialog grant #SA-AUT-2024-015b from Research Corporation for Science Advancement and Arnold and Mabel Beckman Foundation and the U.S. Department of Energy, Office of Science, Energy Earthshot initiatives as part of the Center for Ionomer-based Water Electrolysis at Lawrence Berkeley National Laboratory under Award Number DE-AC02-05CH11231. We thank Ishan Amin for the initial idea and discussion around the LAE encoding. We thank Tobias Kreiman, Sam Blau, Ryan Liu, Ritwik Gupta, Sanjeev Raja, Nithin Chalapathi, Yue Jian, Yiheng Du, Michael Psenka, Boyu Qie, Kai Nelson, Rasmus Hoegh from UC Berkeley / LBNL for the helpful discussions. We thank C. Lawrence Zitnick, Misko Dzamba, Meng Gao, Muhammed Shuaibi, Xiang Fu, Daniel S. Levine, Luis Barroso-Luque and others from the FAIR Chemistry group for helpful discussions and comments.

## Impact Statement

This paper presents work whose goal is to advance the field of machine learning. There are many potential societal consequences of our work, none of which we feel must be specifically highlighted here.

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

# A  Additional OMol25 Results

## A.1  Efficiency

We report the runtime breakdown by component vs. system size in Fig. 8. This profile explains the slope change observed in throughput vs. size curve. For modest N, compute scales like $\mathcal{O}(Nk)$ and is driven by neighborhood attention. Beyond a crossover (earlier for 180M, later for 35M), the all-to-all node attention $\mathcal{O}(N^2)$ component dominates. We note that the profiler adds time and memory overheads and we need the disable `torch.compile`, thus this could differ from the actual performance.

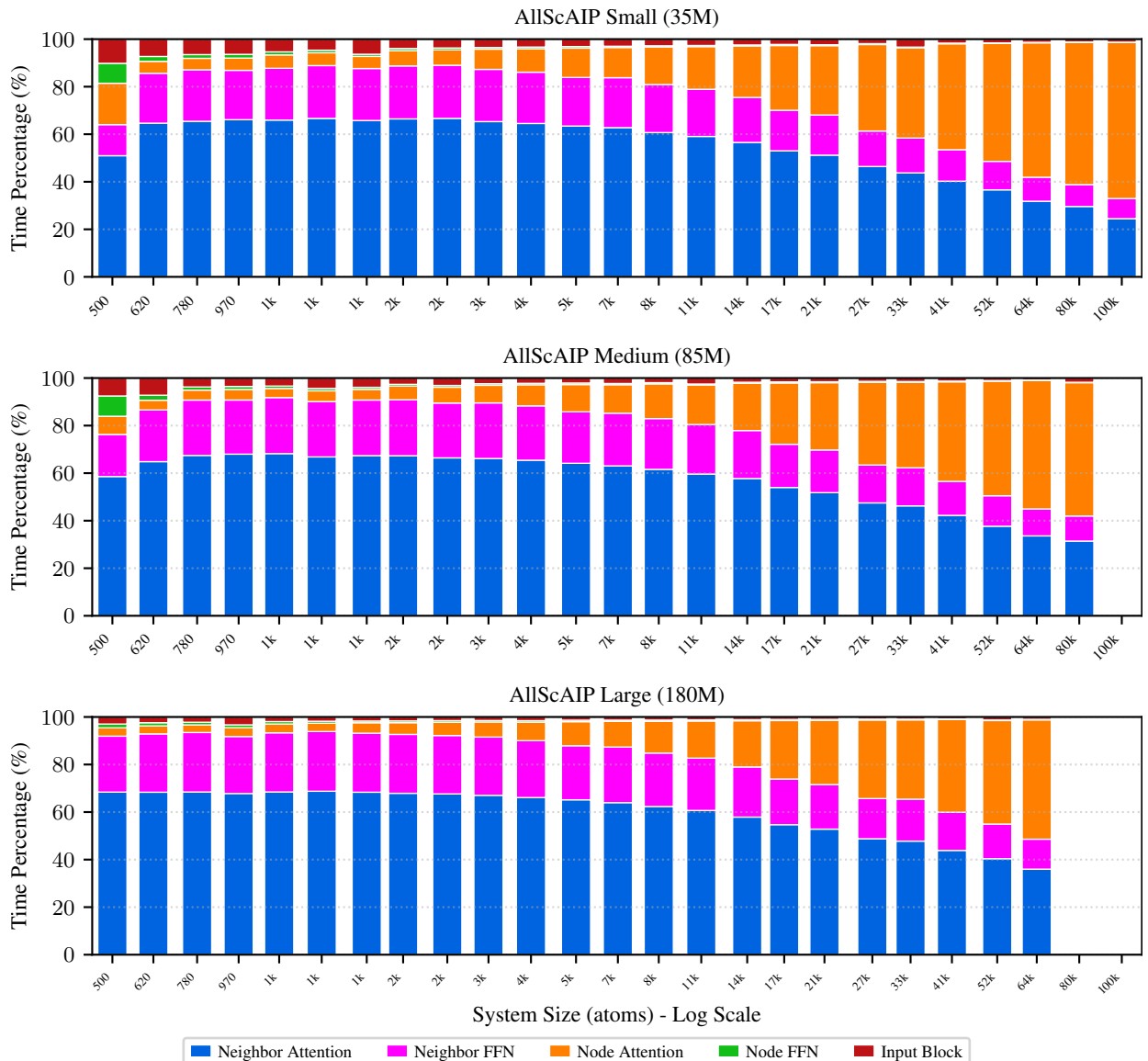

Figure 8. **Runtime breakdown by component vs. system size.** Stacked bars show the percentage of wall-clock inference time spent in each module of AllScAIP on a single H200 (graph construction off), for three model sizes (top: 35M, middle: 85M, bottom: 180M). Colors: Neighbor attention (blue, $\mathcal{O}(Nk)$), Neighbor FFN (magenta), Node attention (orange, $\mathcal{O}(N^2)$), Node FFN (green), and Input/embeddings (red). At small systems the runtime is dominated by the local block (neighbor attention). As the number of atoms grows into the tens of thousands, the cost of the global node attention increases and eventually becomes the majority of runtime; the crossover happens earlier for larger models.

We also provide a raw throughput vs. system size plot for additional comparison (Fig. 9):

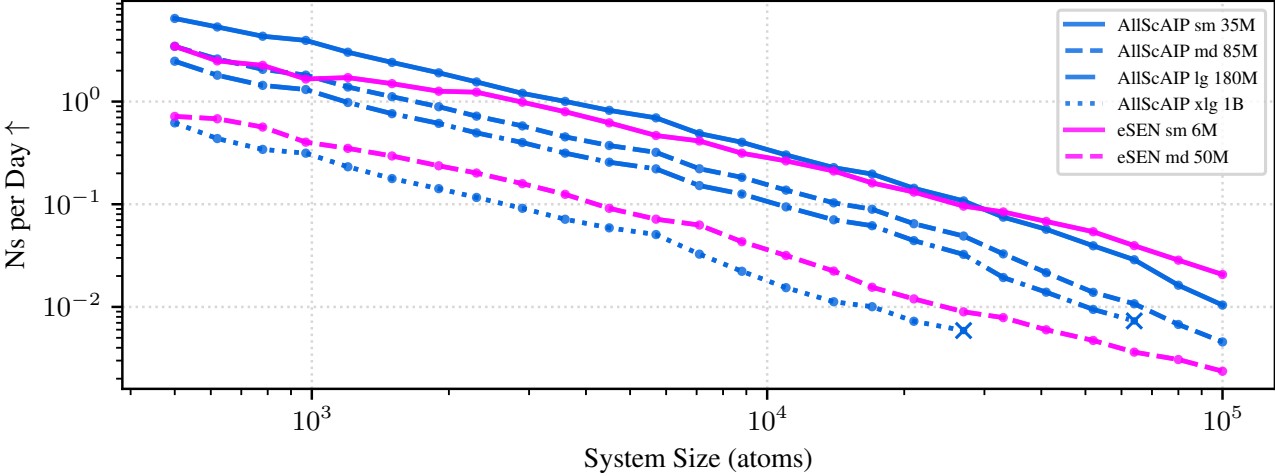

*Figure 9.* **Throughput vs. System size.** Throughput (ns/day, higher is better) is measured on a single H200 141G with graph generation off. Lines show four model sizes (35M/85M/180M/1B) of AllScAIP and eSEN baselines.

## A.2 Long-range Validations on OMol25

To further investigate the long-range error for different models, we bin the OMol25 validation set by system diameter and total charge. We observe that AllScAIP's energy and force MAE remains lower, whereas eSEN's error grows significantly for large-diameter and highly charged systems (Fig. 10).

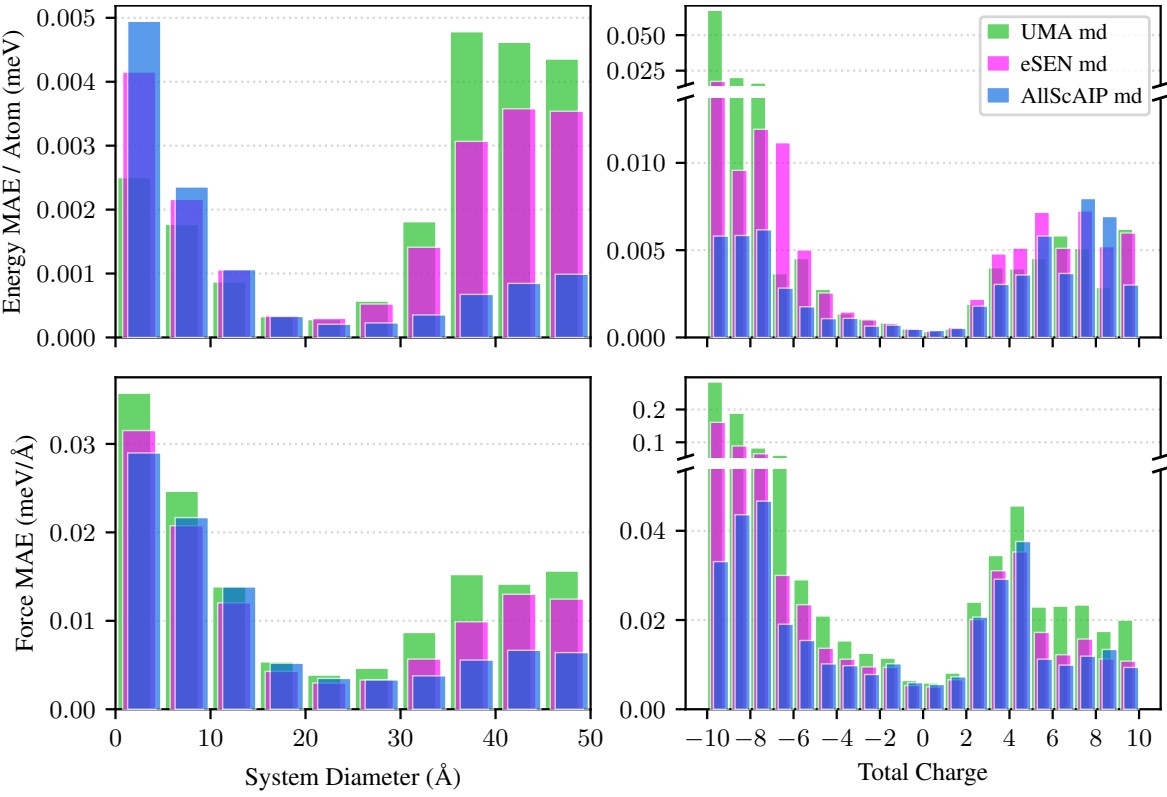

*Figure 10.* **OMol25 Prediction Error vs. System Diameter (Å) and Total Charge.** Energy / Atom MAE (meV, ↓, top) and Force MAE (meV/Å, ↓, bottom) binned by (left) system diameter (Å) and (right) total charge on all of OMol25 validation set. Bars compare AllScAIP-md, UMA-md, and eSEN-md. AllScAIP remains lower and more stable across bins, while eSEN and UMA error grows for large-diameter systems and for high total charge.

## A.3  Visualization of All-to-all Node Attention

To make the long-range behavior of the all-to-all node attention concrete, we visualize one representative large biomolecule (Fig. 10, top-left) and extract attention weights from the final node-attention layer.

**Settings.** For the chosen structure we compute the pairwise distances $r_{ij}$ and collect the post-softmax attention matrices $A^{(h)} \in \mathbb{R}^{N \times N}$ for each head $h$. Distances are binned with $\Delta r = 0.5$Åover $[0,25]$Å.

**Mean attention vs. distance.** For each head we average the attention in each distance bin (row-wise mean, then averaged across rows) and plot the resulting curve. Fig. 10 (top-right) shows all heads (thin) and the head average (thick). A vertical line indicates the local neighbor cutoff used to build the radius graph (6Å). We consistently observe that some heads place above-baseline (1/N) mass at large separations, while others remain local, yielding a flat head-average close to the uniform level.

**Distance and attention heatmaps.** The pairwise distance matrix confirms the contact structure of the system, while the head-averaged attention matrix reveals vertical bands ("hubs"): atoms that receive attention from many others, including at long distances. We fould some hubs coincide with chemically distinctive groups (e.g., charged or polar sites).

**Interpretation.** Taken together, the per-head distance profiles and the heatmaps indicate head specialization and long-range mixing in the node attention layer: a subset of heads routes information across tens of Å, complementing the local neighborhood attention.

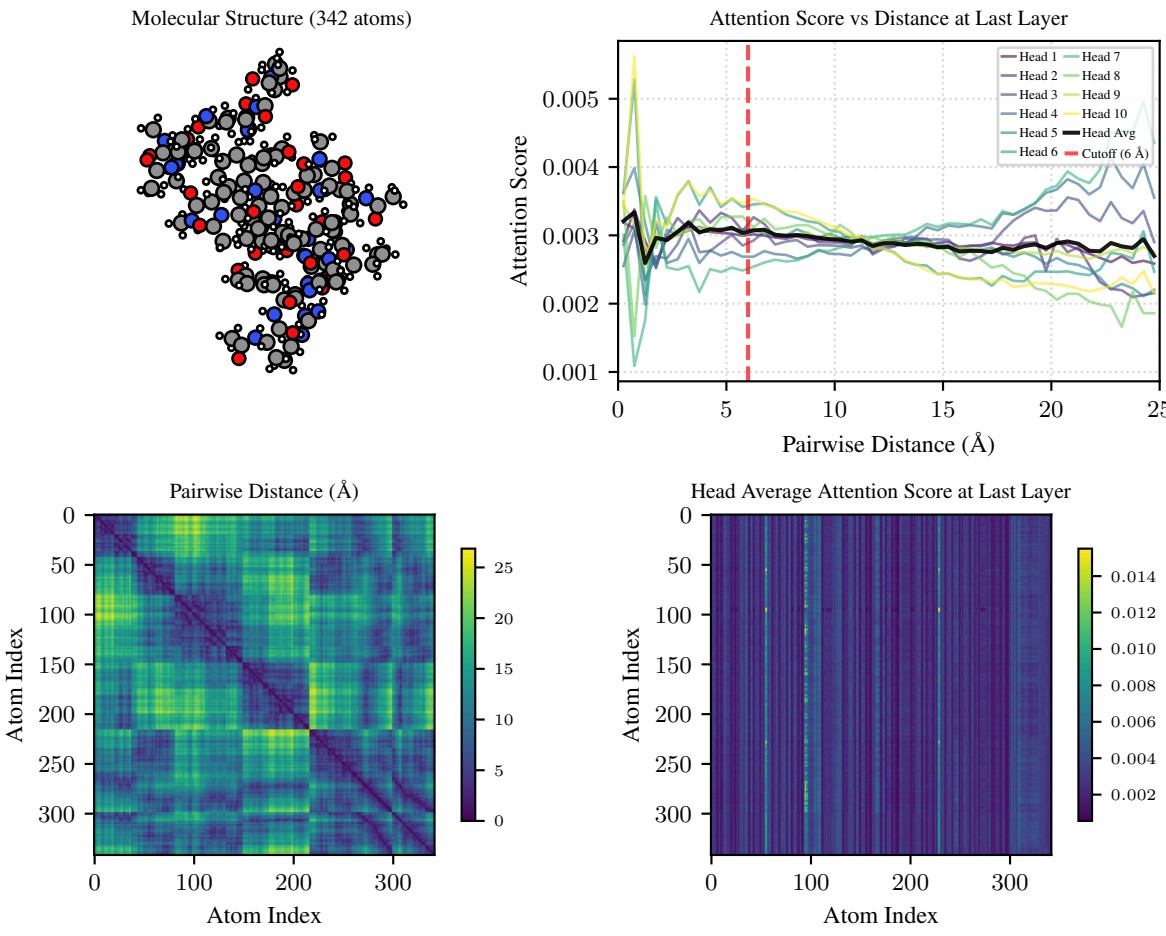

*Figure 11.* **Interpreting the All-to-all Node Attention.** Example protein pocket with 342 atoms in OMol Validation Set (top-left). Top-right: per-head attention weight in the last node-attention layer as a function of pairwise distance; thin lines = individual heads, thick black = head average. The red dashed line marks the local neighbor cutoff (6 Å). Bottom-left: pairwise distance matrix. Bottom-right: head-averaged attention matrix. Several heads allocate noticeable mass to pairs well beyond the local cutoff, and the attention heatmap shows column "hubs" that attract long-range attention across many rows.

*Table 6.* **Component ablations under data and model size scaling**. We report Total Energy MAE (meV) and Force MAE (meV/Å), lower is better. Each size's **with encoding** (Nei Att + LAE + Node Att + ERoPE) configuration is the reference; other rows are colorized relative to that reference (Green = better, Red = worse). The Throughput column reports inference speed (ns/day on one H200 with 1000 atoms). We report results for the OMol25 4M split (80 epochs) and the full 102M (10 epochs).

| | | **Ablations** | | | | **OMol 4M (80 Epochs)** | | | | | | | | | |
| | | | | | | Biomol. | | Elytes. | | Metal Cplx. | | Neutral Org. | | Total | |
| Size | Thrpt | NeiAtt | LAE | NodeAtt | ERoPE | E↓ | F↓ | E↓ | F↓ | E↓ | F↓ | E↓ | F↓ | E↓ | F↓ |
|---|---|---|---|---|---|---|---|---|---|---|---|---|---|---|---|
| 35M | 2.279 | ✓ | ✓ | ✓ | ✓ | 56.9 | 5.57 | 89.3 | 8.93 | 166.2 | 33.52 | 50.3 | 13.52 | 85.2 | 9.35 |
| | 7.623 | ✓ | | ✓ | | 73 | 6.32 | 112.3 | 9.81 | 174 | 34.56 | 58.2 | 14.51 | 102.9 | 10.21 |
| 85M | 1.124 | ✓ | ✓ | ✓ | ✓ | 34.1 | 3.51 | 62.2 | 6.38 | 139.6 | 27.83 | 31.4 | 8.93 | 59.7 | 6.88 |
| | 3.333 | ✓ | | ✓ | ✓ | 43.6 | 4.27 | 70.8 | 7.22 | 153.7 | 29.79 | 40.5 | 10.53 | 68.1 | 7.6 |
| | 1.392 | ✓ | ✓ | ✓ | | 43.9 | 3.78 | 74.2 | 6.67 | 145.2 | 28.26 | 36.4 | 9.03 | 70.1 | 7.1 |
| | 4.014 | ✓ | | ✓ | | 48.3 | 4.24 | 78.4 | 7.16 | 150.6 | 29.19 | 40.8 | 10.12 | 75.2 | 7.55 |
| | 1.281 | ✓ | ✓ | | | 85.8 | 4.58 | 92.9 | 7.42 | 158.6 | 29.86 | 38.8 | 10.01 | 90.3 | 7.93 |
| | 4.327 | ✓ | | | | 90.4 | 4.96 | 98.7 | 7.92 | 164.4 | 30.78 | 42.7 | 10.77 | 95.4 | 8.48 |
| Size | Thrpt | NeiAtt | LAE | NodeAtt | ERoPE | **OMol 102M (10 Epochs)** | | | | | | | | | |
| 35M | 2.279 | ✓ | ✓ | ✓ | ✓ | 33.2 | 3.85 | 48.3 | 6.01 | 122.3 | 26.02 | 25.8 | 8.54 | 48.6 | 6.61 |
| | 7.623 | ✓ | | ✓ | | 43 | 4.47 | 59.2 | 6.73 | 124 | 26.86 | 29.3 | 9.17 | 57.9 | 7.22 |
| 85M | 1.124 | ✓ | ✓ | ✓ | ✓ | 23.7 | 2.81 | 36.3 | 4.59 | 105.3 | 21.93 | 19.7 | 6.03 | 37.3 | 5.1 |
| | 3.333 | ✓ | | ✓ | ✓ | 30.8 | 3.16 | 41.8 | 4.99 | 106.4 | 22.48 | 21.5 | 6.47 | 42.7 | 5.51 |
| | 1.392 | ✓ | ✓ | ✓ | | 30.2 | 2.93 | 41 | 4.76 | 104.3 | 21.59 | 20.5 | 5.94 | 41.7 | 5.13 |
| | 4.014 | ✓ | | ✓ | | 22.4 | 2.91 | 34.6 | 4.7 | 105.7 | 22.31 | 18.7 | 6.29 | 36.1 | 5.23 |
| | 4.327 | ✓ | | | | 63.9 | 3.83 | 58.4 | 5.57 | 116.2 | 23.91 | 22.9 | 7.05 | 60.4 | 6.15 |

*Table 7.* **OMol25 validation results.** Total Energy MAE (meV) and Force MAE (meV/Å) across Biomolecules, Electrolytes, Metal Complexes, and Neutral Organics.

| | | Biomol. | | Elytes. | | Metal Cplx. | | Neutral Org. | | Total | |
| Dataset | Model | E↓ | F↓ | E↓ | F↓ | E↓ | F↓ | E↓ | F↓ | E↓ | F↓ |
|---|---|---|---|---|---|---|---|---|---|---|---|
| All | eSEN-sm-d. | 96.92 | 6.30 | 88.20 | 9.41 | 145.45 | 33.08 | 34.24 | 13.84 | 89.34 | 9.92 |
| | eSEN-sm-cons. | 86.72 | 4.61 | 72.51 | 8.07 | 132.71 | 28.86 | 22.98 | 11.11 | 75.98 | 8.25 |
| | eSEN-md-d. | 50.72 | **2.61** | 47.37 | **4.39** | **97.55** | **19.99** | 16.47 | **5.62** | 49.21 | **4.76** |
| | GemNet-OC | 23.35 | 3.88 | **30.73** | 5.98 | 103.10 | 25.12 | 21.22 | 10.38 | **33.85** | 6.52 |
| | AllScAIP-sm-ft-cons. | 31.94 | 3.34 | 39.70 | 6.06 | 123.69 | 25.94 | 18.77 | 8.97 | 42.48 | 6.41 |
| | AllScAIP-md-d. | **22.25** | 2.92 | 33.68 | 4.84 | 105.59 | 22.36 | 18.57 | 6.30 | 35.18 | 5.25 |
| | AllScAIP-md-cons. | 35.86 | 3.23 | 37.59 | 5.46 | 107.92 | 22.64 | **15.74** | 7.52 | 40.63 | 5.79 |
| 4M | eSEN-sm-d. | 127.43 | 8.12 | 134.78 | 12.64 | 192.77 | 40.44 | 59.84 | 20.17 | 129.77 | 13.01 |
| | eSEN-sm-cons. | 125.62 | 6.17 | 116.65 | 11.16 | 156.48 | 35.33 | 42.00 | 16.92 | 114.36 | 11.09 |
| | eSEN-md-d. | 69.98 | 3.38 | 77.82 | **6.50** | 142.48 | **27.31** | 33.27 | **9.26** | 76.56 | **6.78** |
| | GemNet-OC | 39.58 | 5.20 | 56.32 | 8.42 | 148.50 | 32.76 | 40.98 | 15.59 | 57.82 | 8.98 |
| | AllScAIP-sm-ft-cons. | 43.82 | 4.08 | 62.09 | 7.98 | 130.94 | 32.24 | 29.35 | 12.72 | 59.93 | 8.26 |
| | AllScAIP-md-ft-cons. | **28.27** | **3.62** | **43.85** | 7.21 | **123.35** | 30.68 | **23.11** | 10.54 | **44.40** | 7.51 |

## A.4 Total Energy MAE Results

We present the ablation (Tab. 6 and validation error (Tab. 7) in the total energy MAE metric.

## A.5 Learned Non-PBC Extensivity

As noted in §5.2, the all-to-all node attention does not enforce non-PBC extensivity by construction. Instead, we rely on a ReZero residual (Bachlechner et al., 2021) on the node-attention block (i.e., $x \leftarrow x + \alpha \cdot \text{NodeAtt}(x)$ with $\alpha$ initialized to 0): at initialization the node attention contributes nothing, and the model gradually learns to use it during training. Whether extensivity holds in practice is therefore a property of the *trained* model.

To probe the trained behavior, we place two water molecules at separations from 1.5 to 1,000 Å and compute the interaction energy $|E(2H_2O) - 2E(H_2O)|$ using AllScAIP-md and UMA-md, with ORCA at the $\omega$B97M-V/def2-TZVPD level as ground truth (Fig. 12, left). Both AllScAIP and UMA closely track the DFT reference until they converge to a similar precision floor. The key difference is in the intermediate regime ($\sim$6–30 Å): UMA flatlines immediately past its 6 Å radius cutoff, while

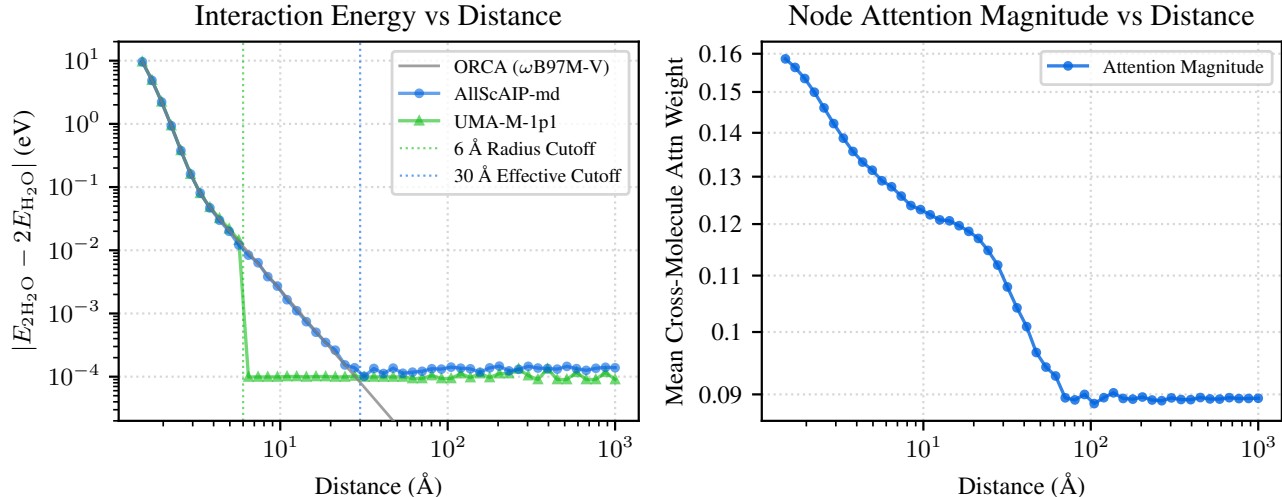

*Figure 12.* **Learned non-PBC extensivity in AllScAIP.** Left: two-water interaction energy $|E(2H_2O) - 2E(H_2O)|$ vs. separation for AllScAIP-md and UMA-md, compared with the DFT reference ($\omega$B97M-V/def2-TZVPD). AllScAIP follows the DFT curve past the 6 Å local cutoff and out to $\sim$30 Å before reaching a precision floor; UMA flatlines immediately past its cutoff. Right: cross-molecule node-attention weights as a function of distance, which decay monotonically and plateau to a uniform floor—the mechanism behind the learned long-range down-weighting.

*Table 8.* OC20 validation (Total Energy) and OMat24 validation results.

| Model | OC20 (Total Energy) | | | | | | OMat24 | | | |
| | Val ID | | | Val OOD-Both | | | Val | | | |
| | Energy | Force | F. Cos. | Energy | Force | F. Cos. | E./Atom | Forces | Stress | F. Cos. |
|---|---|---|---|---|---|---|---|---|---|---|
| UMA-S | 63.6 | 24.1 | 0.63 | 107.0 | 29.2 | 0.65 | 11.3 | 57.1 | 2.9 | 0.98 |
| UMA-M | 43.1 | 15.8 | 0.73 | 70.0 | 19.2 | 0.75 | 10.0 | 47.3 | 2.7 | 0.99 |
| UMA-L | 32.6 | 12.0 | 0.77 | 49.8 | 14.5 | 0.79 | 9.7 | 43.5 | 2.5 | 0.99 |
| eqV2-S | | | | | | | 11 | 49.2 | 2.4 | 0.985 |
| eqV2-M | | | | | | | 10 | 44.8 | 2.3 | 0.986 |
| eqV2-L | | | | | | | 9.6 | 43.1 | 2.3 | 0.987 |
| eSEN | | | | | | | 10.7 | 47.3 | 2.6 | 0.99 |
| AllScAIP-sm-d. | 61.1 | 18.3 | 0.67 | 92.8 | 22.9 | 0.68 | 12.7 | 60.1 | - | 0.98 |
| AllScAIP-sm-ft-cons. | 72.3 | 22.5 | 0.64 | 120.1 | 27.3 | 0.64 | 12.4 | 56.0 | 2.8 | 0.98 |
| AllScAIP-md-d. | 59.3 | 17.6 | 0.69 | 92.2 | 21.8 | 0.70 | 11.4 | 56.6 | - | 0.98 |
| AllScAIP-md-ft-cons. | | | | | | | 10.7 | 54.3 | 2.7 | 0.98 |

AllScAIP continues to follow the DFT curve out to roughly 30 Å. The smooth transition between the neighborhood-attention cutoff and the node-attention-only region indicates that (i) the differentiable kNN handles the boundary smoothly and (ii) the all-to-all node attention learns physically meaningful long-range behavior.

Fig. 12 (right) shows the cross-molecule node-attention weights as a function of distance: they decay monotonically and plateau to a uniform precision floor. This is the mechanism behind the learned extensivity: the model down-weights distant atoms, so the contribution from a far-away copy of the system effectively vanishes. Together with the near-perfect non-PBC additivity reported in Tab. 3 and the $\sim$10$\times$ improvement over UMA on the OMol25 distance-scaling evaluation (Fig. 6), this confirms that AllScAIP learns non-PBC extensivity reliably in practice.

# B  OC20 and OMat24 Results

**Settings.** We train AllScAIP on OC20 (240M) and OMat24 (100M) datasets. On OC20, we first train direct force for 3 epochs, and then conservative fine-tune for 1 epoch. We enable fp16 autocast during direct force training. On OMat24, we do 6 epochs direct force + 3 epochs conservative fine-tune, and it is full fp32 training.

*Table 9.* **SPICE test results.** Energy / Atom MAE (meV, ↓) and Force MAE (meV/Å, ↓) across different splits of SPICE. AllScAIP achieve the lowest overall energy and force errors.

| Dataset | MACE-L E | MACE-L F | MACE-M E | MACE-M F | MACELES-M E | MACELES-M F | EScAIP E | EScAIP F | eSEN E | eSEN F | AllScAIP E | AllScAIP F |
|---|---|---|---|---|---|---|---|---|---|---|---|---|
| PubChem | 0.88 | 14.75 | 0.91 | 20.57 | 0.71 | 17.24 | 0.53 | 5.86 | 0.15 | 4.21 | **0.13** | **3.21** |
| DES370K M. | 0.59 | 6.58 | 0.63 | 9.36 | 0.54 | 7.65 | 0.41 | 3.48 | 0.13 | 1.24 | **0.07** | **0.97** |
| DES370K D. | 0.54 | 6.62 | 0.58 | 9.02 | 0.47 | 7.48 | 0.38 | 2.18 | 0.15 | 2.12 | **0.06** | **0.95** |
| Dipeptides | 0.42 | 10.19 | 0.52 | 14.27 | 0.52 | 11.46 | 0.31 | 5.12 | 0.25 | 3.68 | **0.06** | **1.54** |
| Solvated A.A. | 0.98 | 19.43 | 1.21 | 23.26 | 0.82 | 18.37 | 0.61 | 11.52 | 0.25 | **3.68** | **0.13** | 4.86 |
| Water | 0.83 | 13.57 | 0.76 | 15.27 | 0.69 | 12.54 | 0.72 | 10.31 | 0.15 | 2.50 | **0.08** | **2.12** |
| QMugs | 0.45 | 16.93 | 0.69 | 23.58 | 0.76 | 20.11 | 0.41 | 8.74 | 0.12 | 3.78 | **0.11** | **2.73** |

*Table 10.* **Differences between EScAIP and AllScAIP.** "Same" indicates the component is unchanged from EScAIP.

| Component | EScAIP (Qu & Krishnapriyan, 2024) | AllScAIP (this work) |
|---|---|---|
| Neighborhood self-attention | Fixed-radius graph, MHSA over neighbors | Same (with optional LAE) |
| Angular encoding | — | **Legendre Angular Encoding (LAE)** on Q/K in neighborhood attention |
| Global / long-range stage | — | **All-to-all node self-attention** over all atoms per graph |
| Radial encoding (global) | — | **Euclidean Rotary Position Encoding (ERoPE)** on node-attention logits |
| Graph construction | Fixed-radius graph | **Differentiable kNN** graph (Liu et al., 2026) (enables energy conservation) |
| Force prediction | Direct force head | Direct force *or* energy-gradient (conservative) options |

**Results.** We evaluate on OC20 using the Total Energy MAE (ID and OOD-Both) on the validation set, and OMat24 validation set (Tab. 8). We comparing with UMA (Wood et al., 2025), which is trained on more data (459M) and has a larger compute budget. On OC20, the medium model (AllScAIP-md-d.) shows competitive accuracy. It surpasses UMA-S in both Val/ID and Val/OOD-Both. The OOD/ID energy ratio ($\approx$1.6) is on par with UMA-M, indicating healthy generalization. On OMat24, the medium conservative model shows better performance than UMA-S. We did not devote substantial effort to model tuning for these two datasets.

# C    SPICE Results

**Settings.** To compare with other long-range MLIPs (e.g. LES (Kim et al., 2025)), we trained a small (34M), energy-conserving variant (`AllScAIP-sm-cons.`) on the SPICE dataset using the MACE-OFF splits (Eastman et al., 2023), following Kim et al. (2025) (we note that the SPICE dataset is included and recalculated at a higher level of theory in OMol25 (Levine et al., 2025)). We train for 300 epochs on H200 142GB with full fp32 precision.

**Results.** We evaluate on seven held-out test subsets PubChem, DES370K monomers/dimers, Dipeptides, Solvated Amino Acids, Water, and QMugs. We report Energy/Atom MAE (meV; lower is better) and Force MAE (meV/Å; lower is better). Tab. 9 compares against MACE/MACELES (Kim et al., 2025), EScAIP (Qu & Krishnapriyan, 2024), and eSEN (Fu et al., 2025). We found that `AllScAIP-sm-cons` achieves the lowest overall energy and force errors across splits. These results indicate that the model recipe is useful in different dataset settings, and it shows superior performance over other long-range MLIP designs, such as MACELES (Kim et al., 2025).

# D    Differences Between AllScAIP and EScAIP

AllScAIP builds on the EScAIP backbone (Qu & Krishnapriyan, 2024) and adds three architectural ingredients on top of it, along with an updated graph construction. In particular, the "neighborhood attention only" ablation in Tab. 2 corresponds to EScAIP. Tab. 10 summarizes the changes.

# E    Additional MD Simulations

To assess MD structural fidelity under distribution shift, we ran *zero-shot* NVT simulations on seven MD22 molecules (Chmiela et al., 2023) using the OMol-trained model (Fig. 13) We computed the radial distribution $h(r)$ from the production segments and compared against reference trajectories generated with PBE+MBD, which a lower level of theory than OMol, hence not expected to coincide with our training target. Across systems, AllScAIP reproduces the same peak positions and overall shapes as eSEN and UMA, indicating comparable equilibrium structure and no gross biases in intermolecular distances.

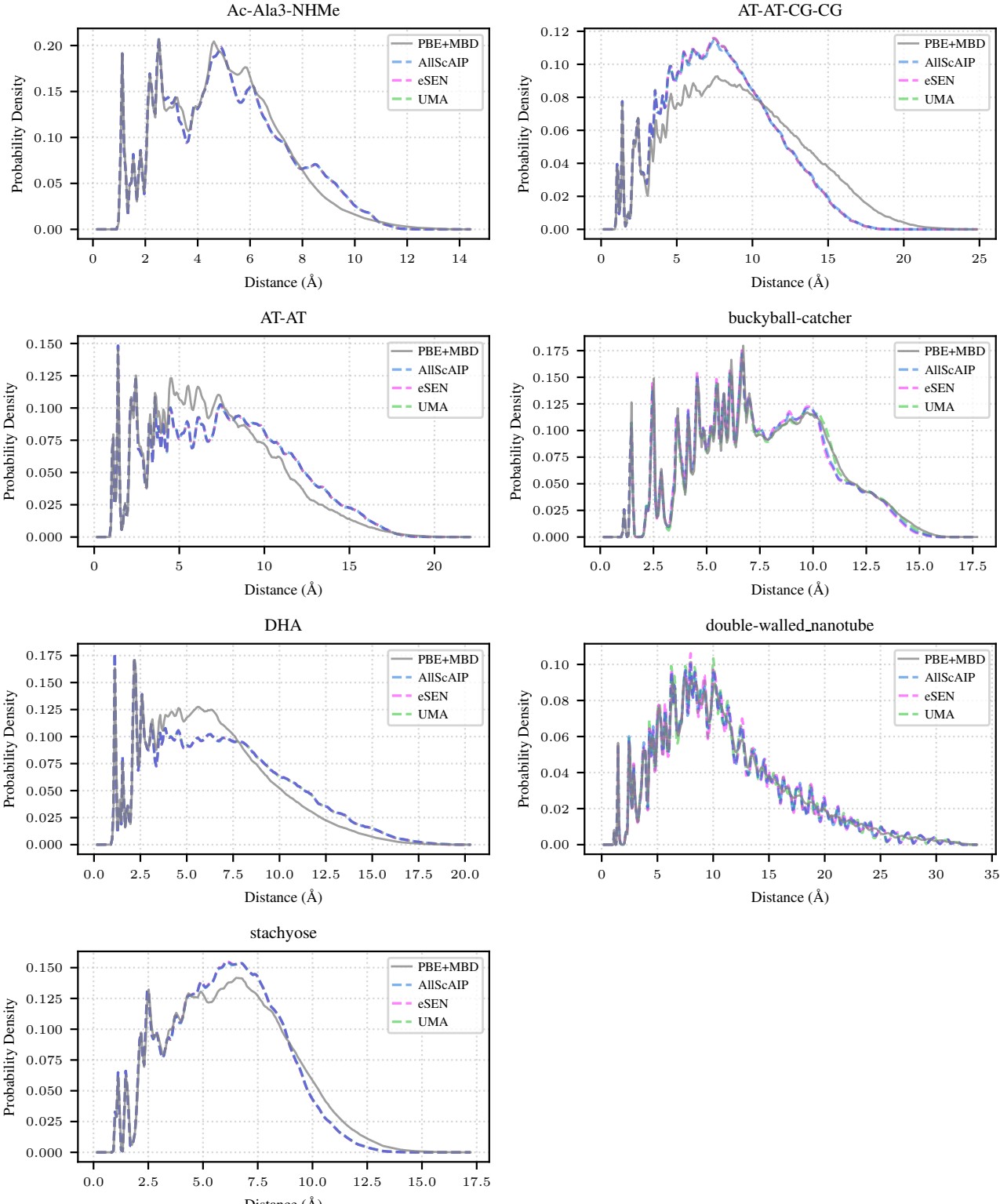

*Figure 13.* **Zero-shot MD on MD22: radial distribution** $h(r)$**.** NVT trajectories with OMol-trained AllScAIP on seven MD22 systems. Curves show $h(r)$ from the simulations compared with PBE+MBD reference runs. Although the reference level differs from OMol, AllScAIP tracks eSEN/UMA in peak locations and overall shape, indicating matched structural statistics.

## F    Detailed Model Configurations

*Table 11.* AllScAIP Model Configurations

| Model Size | small | medium | large | xlarge |
|---|---|---|---|---|
| hidden_size | 512 | 640 | 1024 | 2048 |
| num_layers | 6 | 10 | 8 | 12 |
| atten_num_heads | 8 | 10 | 16 | 32 |
| atten_name | | memory_efficient | | |
| activation | | gelu | | |
| normalization | | rmsnorm | | |
| node_direction_expansion_size | | 10 | | |
| edge_direction_expansion_size | | 6 | | |
| edge_distance_expansion_size | | 512 | | |
| attn_num_freq | | 32 | | |
| ffn_hidden_layer_multiplier | | 2 | | |
| output_hidden_layer_multiplier | | 2 | | |
| max_num_elements | | 110 | | |
| max_batch_size | | 96 | | |
| knn_soft | | True | | |
| knn_sigmoid_scale | | 0.2 | | |
| knn_lse_scale | | 0.1 | | |
| knn_use_low_mem | | True | | |
| distance_function | | gaussian | | |
| use_envelope | | True | | |
| mlp_dropout | | 0.0 | | |
| atten_dropout | | 0.0 | | |

