# OpenReview forum: "A recipe for scalable attention-based ML potentials: unlocking long-range accuracy with all-to-all node attention"
_ICML.cc/2026/Conference — ICML 2026 regular_

### Official Review · Reviewer_yfFy · 2026-03-06

**Soundness:** 3
**Presentation:** 4
**Significance:** 3
**Originality:** 2
**Overall Recommendation:** 4
**Confidence:** 4

**Summary:**

This work introduces an attention-based machine learning force field to handle long-range interactions in molecular compounds. Their AllScAIP called model consists of two attention mechanisms, one local and one global. The local neighborhood attention captures high-frequency angular and distance perturbations while the global attention mechanism captures long-range interactions. The paper concerns itself with the tarde-off between physical inductive biases and learnable expressive parametrizations.

**Compliance With Llm Reviewing Policy:**

Affirmed.

**Final Justification:**

The authors did a great job of highlighting the strengths of their approach. Still, I remain slightly skeptical about this work's real-world impact as forces, the quantity of interest, remain the same.

**Key Questions For Authors:**

**Q1**: The non-PBC size extensivity especially on the untrained model is a very surprising result to me due to the global attention. Is this since we replicate the molecule? In a more general sense, I would expect for the two different molecules M1 and M2, the total energy (when sufficiently apart) to be E(M1) + E(M2). Does this still work?

**Q2**: The results of the MD simulations in Figure 6 are very promising. However, I wonder why eSEN performs worse here despite its lower force error?

**Q3**: For Figure 4, it would be great to see the inclusion of more models like GemNet-OC (since it does not use spherical harmonics but encodes rotation invariance) and in general how this behaves in terms of forces. From Table 4, I'd suspect the front to look different given that eSEN seems to work better on forces?

**Limitations:**

The authors do not directly address limitations of their method. Given the current evaluation, it seems that energy improvements are prioritized. It would be great to know whether these must come at a cost of force predictions. Furthermore, the OC20 and OMat24 results seem to show that AllScAIP is more tailed towards vacuum computations rather than materials.

**Strengths And Weaknesses:**

Strengths:
* The hypothesis that inductive biases matter more in small-scale and data-limited regimes is very reasonable and supported throughout many subfields of deep learning.
* The paper is very well written and easy to follow. The authors do a great job of explaining their design choices to the reader and ablate them well.
* The experiments support the claims well and the authors do a great job of evaluating their method across a series of important downstream applications, i.e., MD simulations, instead of relying on ID prediction errors.

Weaknesses:
* The authors write to LAE that the inner product of spherical harmonics corresponds to a Legendre polynomial of $cos(\theta)$. However, this breaks once one multiplies each element of the $2l+1$ vector with a different scalar as done here.
* The improvements seem to be primarily affect energy which makes sense as long range interactions are low-frequency by definition and affect forces less.
* At times, the strong focus on energy and the low variety of reference models make clear judgements more difficult. For instance, in practice, force predictions matter a lot more than energy predictions as these are important in MD simulations.
* See questions.

---

> ### Author Rebuttal · Authors · 2026-03-31
>
> We thank the reviewer for the thoughtful comments.
>
> **W1 (LAE inner product with different scalars)**
>
> The reviewer is correct that elementwise modulation by Q/K projections means the dot product is no longer a single $P\_l(\cos \theta)$, but a learned weighted sum of Legendre polynomials across degrees. This is intentional: it lets each head learn its own angular frequency weighting. Rotation invariance of the logits is preserved because the same modulation is applied to both Q and K within each degree block. We use the LAE name since the encoding is still built from Legendre polynomials via the addition theorem. We will clarify this in the revision.
>
> **W2 (Improvements mainly affect energy)**
>
> This is partly expected: long-range interactions are lower-frequency and contribute more to total energy than to per-atom force vectors. But there is also a compute gap to consider. AllScAIP is faster per training step than eSEN at comparable sizes, and we match data budgets (same splits, same epochs), not compute budgets. At the 102M setting AllScAIP sees the same data with less total compute. Despite this, AllScAIP achieves better energy across all splits. The force gap may narrow with additional training or loss weight tuning.
>
> **W3 + Q2 (Force predictions and MD quality)**
>
> Lower validation force MAE does not always reflect on better MD quality [1,2]. The validation set is dominated by small molecules where long-range effects are weak, so eSEN's lower force MAE there does not transfer to large-scale MD. In our MD on 39 molecular liquids with thousands of atoms (Sec. 5.3, Fig. 6), AllScAIP produces densities and heats of vaporization closer to experiment than eSEN. eSEN systematically over-compresses liquids, which points to missing long-range interactions. AllScAIP's node attention mitigates this, producing more physical bulk behavior even with slightly higher validation force MAE.
>
> **Q1 (Size extensivity with global attention)**
>
> Yes, this should work for two different molecules M1 and M2 placed far apart. We observe that the learned attention scores decay approximately as 1/r with distance and approach zero at large separations. In our extensivity test, we place the two copies at 1000 Angstrom separation, which is far enough that the inter-molecule attention is effectively zero. We will add a plot of this in the revision to make this decay behavior explicit.
>
> **Q3 (Including GemNet-OC and force-based Pareto)**
>
> Unfortunately, GemNet-OC is not supported in the current codebase and its checkpoints are not publicly released, so we cannot reproduce a fair speed comparison on the same hardware.
>
> **References**
>
> [1] ​​Forces are not Enough: Benchmark and Critical Evaluation for Machine Learning Force Fields with Molecular Simulations https://arxiv.org/abs/2210.07237
>
> [2] The dark side of the forces: assessing non-conservative force models for atomistic machine learning https://arxiv.org/abs/2412.11569

---

> > ### Author Rebuttal · Reviewer_yfFy · 2026-04-01
> >
> > I would like to thank the authors for their response. They support my score. Regarding Q1, it would be great to see empirical data on this with different molecules as I remain skeptical.

---

> > > ### Author Response · Authors · 2026-04-04
> > >
> > > We thank the reviewer for the follow-up and for pressing on this point. The reviewer is correct: non-PBC extensivity in AllScAIP is not enforced by construction, but learned. The reason our original test (Table 3) showed near-perfect extensivity is that we use ReZero residual connections for the node attention (x = x + $\alpha$ * node_att(x), with $\alpha$ initialized to 0). At initialization, the node attention contributes nothing, and the model gradually learns to use it during training. The extensivity behavior is therefore a property of the trained model, not a hard architectural guarantee. We will correct this in the revision.
> > >
> > > That said, we have conducted two additional experiments showing that the model does learn extensivity reliably.
> > >
> > > **1. Size extensivity with different molecules.** We randomly sampled 10,000 pairs of distinct molecules (M1, M2) from the OMol validation set and placed them 1,000 Angstrom apart in vacuum (no PBC). We report the per-atom Energy Difference |E(M1+M2) - E(M1) - E(M2)| / N:
> > >
> > > | Model | Same system (paper) | Different system (new) |
> > > |---|---|---|
> > > | UMA-M-1.1 | $3.83\times 10^{-3}$ meV | $3.61\times 10^{-3}$ meV |
> > > | AllScAIP-md | $2.49\times 10^{-3}$ meV | $2.84\times 10^{-3}$ meV |
> > >
> > > Both models achieve comparable errors in the same-system and different-system settings, confirming that extensivity holds in practice for arbitrary molecule pairs at large separation, not just replicated systems.
> > >
> > > **2. Distance-resolved interaction energy and attention decay.** ([Figure Link](https://doi.org/10.5281/zenodo.19413417)) We placed two water molecules at distances from 1.5 to 1,000 Angstrom and computed the interaction energy |E(2H2O) - 2E(H2O)| using AllScAIP, UMA, and ORCA (wB97M-V/def2-TZVPD) as ground truth (left figure). Both AllScAIP and UMA closely track the DFT reference until converging to a similar precision floor. The key difference is in the intermediate regime (6-30 Angstrom): UMA flatlines immediately beyond its 6 Angstrom radius cutoff, while AllScAIP continues to follow the DFT curve till around 30 Angstrom. The smooth transition between the neighborhood attention cutoff and the node-attention-only region shows that (1) the smooth differentiable kNN handles the boundary well and (2) the node attention learns meaningful long-range behavior. We also extract the cross-molecule node attention weights (right figure), which decay monotonically with distance and plateau to a uniform precision floor. This confirms that the model learns to down-weight distant atoms, which is the mechanism behind the learned extensivity.
> > >
> > > **3. Long-range accuracy on OMol benchmarks.** Consistent with the above, AllScAIP achieves roughly 10x lower error than UMA on the distance-scaling task of the OMol evaluation, further confirming that the all-to-all attention learns physically meaningful long-range interactions.

---

### Official Review · Reviewer_2Ypo · 2026-03-09

**Soundness:** 3
**Presentation:** 3
**Significance:** 3
**Originality:** 3
**Overall Recommendation:** 5
**Confidence:** 2

**Summary:**

This paper proposes AllScAIP, an attention-based MLIP designed for scalable long-range modeling. The model combines two attention stages: a local neighborhood attention layer for short-range interactions and an all-to-all node attention layer for modelling global interactions. It also includes two lightweight geometric priors, LAE for local angular information and ERoPE for distance-aware global encoding. The main claim is that some inductive biases, such as rotational equivariance and aspects of long-range modeling, can be learned under scale, while global node attention remains essential for strong performance. The paper evaluates the method on several benchmarks, with the strongest results on OMol25 and additional results on SPICE, OMat24, and OC20.

**Compliance With Llm Reviewing Policy:**

Affirmed.

**Final Justification:**

All concerns have been addressed. I recommend an acceptance.

**Key Questions For Authors:**

1) On the global node attention: The paper says global node attention enables many-hop communication in one step. Then whether a deep GNN stack still necessary? Please discuss more clearly whether global node attention can replace part of the role of deeper local message passing. The comparison between “deeper model without global attention” and “shallower model with global attention.” can help show whether the two (global node attention and multiple GNN interaction blocks) are complementary or partially interchangeable.

2) On the results of Table 4: In Table 4, AllScAIP is better than eSEN on the 4M setting, but not clearly better on the full dataset. Please explain this more carefully. Why does the advantage degrade after using the full dataset? Is it necessary to further scale the parameter size of the AllScAIP model? If so, roughly how large should the model be scaled in order to reach a prediction accuracy comparable to eSEN?

**Limitations:**

This is a strong and thoughtful paper. The main idea is clear, the model is well designed, and the experiments are extensive. My main concern is the quadratic cost of all-to-all node attention and the fact that the strongest gains appear mainly on molecular long-range benchmarks rather than uniformly across all domains.

**Strengths And Weaknesses:**

Strengths
1) The paper studies which inductive biases remain necessary when model and dataset scale increase. This is a meaningful question for the future design of large MLIPs.
2) The model design is clean. The combination of local attention and global node attention is easy to understand. The paper does not rely on many heavy physical modules, which makes the architecture relatively simple.
3) The ablation study is useful. The paper clearly shows different roles for LAE, ERoPE, and global node attention. In particular, the results suggest that the benefit of LAE and ERoPE decreases with scale, while all-to-all node attention remains important.
4) The evaluation is comprehensive. The paper does not only report energy and force errors. It also studies extensivity, rotational behavior, NVE energy drift, and downstream MD observables such as density and heat of vaporization.

Weaknesses
1) On efficiency investigation. The all-to-all node attention layer has quadratic complexity in the number of atoms. This design may create a practical ceiling for very large systems. This limitation should be discussed more directly such as if the atom number is over 100,000 or larger.
2) On the accuracy on material dataset. On materials and catalyst benchmarks such as OMat24 and OC20, the method looks competitive, but not clearly state of the art.

---

> ### Author Rebuttal · Authors · 2026-03-31
>
> We thank the reviewer for the positive feedback and comments.
>
> **W1 (O(N^2) complexity for large systems)**
>
> We agree the all-to-all node attention is O(N^2) in FLOPs. For systems above 100k atoms, no MLIP currently provides accurate ab-initio-level MD to our knowledge. That said, the O(N^2) attention is not a fundamental architectural barrier. LLMs already handle 1M+ token contexts using the same underlying attention mechanism, with engineering solutions (flash attention, ring attention, etc.) that transfer directly to our setting. Naive radius/kNN graph construction also requires pairwise distance computation (O(N^2)) and faces similar scaling challenges; both problems admit the same class of spatial decomposition and engineering solutions. We see scaling to larger systems as an engineering problem with known approaches, not an architectural limitation.
>
> **W2 (OMat24/OC20 not clearly SOTA)**
>
> We agree. Our OMat24 and OC20 results are transfer sanity checks without extensive tuning, not leaderboard claims. Our primary claims and ablations center on OMol25. We will clarify this scope in the revision.
>
> **Q1 (Global node attention vs. deeper local message passing)**
>
> There is extensive literature on over-smoothing and over-squashing in deep GNNs [1], which shows that simply stacking more message-passing layers does not reliably extend effective receptive fields. As a concrete example, UMA-md uses 10 layers, giving a theoretical receptive field of ~60 Angstroms, yet it still degrades sharply on the distance-scaling evaluation (Fig. 5) while AllScAIP stays flat. This suggests that global attention provides better long-range abilities that depth alone does not.
>
> **Q2 (Table 4: advantage degrades at full dataset)**
>
> There are two factors that could explain this:
>
> First, AllScAIP is faster per training step than eSEN at comparable sizes, so when we match data budgets (same splits, same epochs), AllScAIP uses less total compute. This compute gap grows with dataset size: at 4M we train for 80 epochs, at 102M we train for 10 epochs. Since AllScAIP's per-step speed advantage is constant, the absolute compute savings are larger at 102M (more steps total). This larger compute budget gap could explain the smaller performance gap.
>
> Second, GNN-based models like eSEN reuse parameters across message-passing layers, so raw parameter counts are not directly comparable with transformer-style models that have separate parameters per layer. We believe scaling up AllScAIP's parameters further would widen the gap, but we are limited by compute for this submission.
>
> **References**
>
> [1] A Survey on Oversmoothing in Graph Neural Networks https://arxiv.org/abs/2303.10993

---

> > ### Author Rebuttal · Reviewer_2Ypo · 2026-04-02
> >
> > The response have addressed my issues.

---

### Official Review · Reviewer_7fTr · 2026-03-13

**Soundness:** 2
**Presentation:** 2
**Significance:** 3
**Originality:** 3
**Overall Recommendation:** 4
**Confidence:** 4

**Summary:**

The paper proposes a solution to the long-rang interaction problem that exists in machine learning interatomic potentials (MLIPs). The solution is in the form of a so called "all-to-all attention" layer that allows all atoms/nodes to attend to one another, instead of restricting interaction to some radial cutoff. The resulting model is demonstrated by training on several large DFT datasets, where achieves competitive results with existing methods and state-of-the-art results on downstream evaluations for OMol25.

**Compliance With Llm Reviewing Policy:**

Affirmed.

**Final Justification:**

The rebuttal addressed my concerns regarding overstated claims and inconsistencies in the figures. I have increased my score to Weak Accept on the basis that the evaluations are mainly limited to OMol25 data.

**Key Questions For Authors:**

1. Can the authors detail the exact changes from EScAIP (see weakness 1). This would help improve the clarity of the paper.

2. Better baselines and evaluations for OC and OMat (see weakness 2d and 2e) should be made to claim "competitive" performance, otherwise it seems like these claims are overstated.

3. The remaining evaluations (downstream tasks, pareto curves) mostly focus on energy errors or differences for OMol data, so it is less clear how robust the method actually is. It would be helpful to show how this is not simply a result of e.g. a higher energy weight compared to eSEN or UMA, and/or offer evaluations that are more influenced by force predictions.

4. In figure 10, why is the force error lower for AllScAIP in all systems, while being higher for all systems (except Biomol. with 4M dataset) (see weakness 2c).

**Limitations:**

Yes.

**Strengths And Weaknesses:**

**Strengths**

 1. The proposed method addresses an open issue in MLIPs, especially for organic molecules.
 2. The resulting model outperforms prior work in energy erorrs on the OMol dataset and on downstream OMol25 test evaluations.
 3. The model leverages the standard attention mechanism, which is appealing from an engineering perspective as it is the focus of many ongoing optimization efforts for models in other fields.

**Weaknesses**

 1. It is difficult to determine all of the exact changes are in the proposed work over the EScAIP model in which the authors build off of. The all-to-all node attention, Legendre angular encoding, and rotary position encoding are all mentioned as additions in section 1; however there is also brief mention of a differentiable kNN graph construction method that does not seem to be used in EScAIP. It would be helpful to illustrate all of these changes either in the architecture diagram (e.g. have EScAIP and AllScAIP next to eachother, highlighting new components), or in equations.

 2. There are several overstated claims and inconsistencies in the paper:

       a. It is claimed in the abstract and throughout that the model achieves state-of-the-art energy *and* force errors on OMol25, however it appears that in Table 4 that it consistently under performs for force errors. The remaining evaluations are all focused on energy predictions. This improvement could be due in part to the weighting of energy and force terms during training.

      b. The "locality prior" and "long-range effect" priors seem somewhat contradictory. Does adding the all-to-all attention basically eliminate the locality/smooth cutoff? While the empirical extensivity tests do somewhat address this, it is not clear how extensivity is explicitly enforced while simultaneously allowing for an "infinite" receptive field.

      c. In Figure 10., AllScAIP produces lower force MAE than eSEN for all system diameters and charges, despite showing *higher* force MAE across all splits but one.

      d. The OC results are misleading, as UMA is trained on a different level of theory than OC20. This should not be used as a baseline, and instead be compared to e.g. [1] which does considerably better. The newer OC25 dataset also has baselines for eSEN.

      e. A more standard comparison for OMat24 would be the eSEN models, which have validation errors reported in [2] (table 15) and have comparable errors to UMA despite only being trained on OMat. This negates the claims regarding training on more data. Furthermore, standard benchmarks such as Matbench Discovery or MDR Phonons should be included when comparing MLIP materials models.

       f. While the $O(N^2)$ scaling is indeed in part an engineering problem, several mentioned attention optimizations such as KV-caching and paging only apply to auto-regressive use cases of attention. This would only make sense if e.g. all the atoms were fixed and one was added or moved, which is atypical for most MD simulations.


[1] Liao, Yi-Lun, et al. "Equiformerv2: Improved equivariant transformer for scaling to higher-degree representations." arXiv preprint arXiv:2306.12059 (2023).

[2] Wood, Brandon M., et al. "Uma: A family of universal models for atoms." arXiv preprint arXiv:2506.23971 (2025).

---

> ### Author Rebuttal · Authors · 2026-03-31
>
> We thank the reviewer for the thoughtful comments.
>
> **W1 + Q1 (Changes from EScAIP)**
>
> EScAIP uses only neighborhood self-attention on a fixed radius graph. AllScAIP adds three new components on top: (1) all-to-all node self-attention, (2) Legendre Angular Encoding (LAE), and (3) Euclidean Rotary Position Encoding (ERoPE). In particular, the "neighborhood attention only" ablation in Table 2 is equivalent with EScAIP. The differentiable kNN graph construction comes from a concurrent work [1] and replaces the fixed radius graph used in EScAIP. We will add an explicit table of changes in the revision to make this clearer.
>
> **W2a + Q3 (SOTA claim and force errors)**
>
> We will revise the claim to state-of-the-art energy accuracy, with competitive force errors. The gap is not purely a reweighting artifact. AllScAIP achieves ~10x lower error on the distance-scaling evaluation (Fig. 5, Table 5), where molecules are uniformly stretched/compressed beyond the training distribution. This test probes long-range accuracy directly, and eSEN/UMA degrade sharply while AllScAIP stays flat. The large energy gains on the Biomolecules and Electrolytes splits (Table 4), where long-range effects matter most, further confirm that the improvement comes from better long-range modeling, not just loss weighting.
>
> Additionally, eSEN is slower per step than AllScAIP at comparable model sizes, and our training matches eSEN's data budget (same splits, same epochs), not its compute budget. So AllScAIP uses less total compute and still achieves better energy and competitive forces. With the same compute budget the gap could possibly be closed.
>
> It also has been widely discussed in literature that lower force MAE also does not always translate to better MD simulations [2,3]. Our MD results (on 39 molecular liquids) show that AllScAIP produces densities and heats of vaporization closer to experiment than eSEN, despite eSEN's lower force MAE on the validation set (Sec. 5.3, Fig. 6). AllScAIP removes the systematic over-compression bias seen in eSEN, which we attribute to the long-range interactions captured by the node attention.
>
> **W2b (Locality prior vs. long-range)**
>
> These are not contradictory. The neighborhood attention stage gathers local information through a kNN graph, which provides the locality prior. The node attention stage then operates on the full set of atoms for global communication. The alternative (that would lack a locality prior) would be removing the graph construction entirely and using only node attention. Our two-stage design is deliberate: local attention handles fine geometry and anisotropic short-range chemistry, while node attention adds long-range coupling on top. Extensivity is maintained by construction (sum aggregation for energy, PBC minimal image distance for graph construction and node attention), and we verify this numerically in Table 3.
>
> **W2c + Q4 (Figure 10 inconsistency)**
>
> We thank the reviewer for pointing this out. There was a bug in the original Figure 10 that caused the inconsistency, and we have fixed it; the updated plot is at https://doi.org/10.5281/zenodo.19341882 We will update to the correct figure in the revisions.
>
> **W2d (OC20 baselines)**
>
> We train on the same OC20++ total energy targets that UMA uses, which differ from the original OC20 S2EF/IS2RE setup used by eqV2 and the public leaderboard. A direct comparison against eqV2 would mix protocols and training target, which is why we compare against UMA under matched conditions. We will clarify this distinction in the revision.
>
> **W2e (OMat24 baselines)**
>
> We will add eSEN numbers to the OMat24 table. Our energy errors are still competitive. We note that our OMat24 and OC20 results are intended as transfer checks (training on these datasets without extensive tuning), not as primary claims. Our main scientific claims and ablations center on OMol25. Matbench Discovery uses an older pseudopotential than OMat24, so a model trained on OMat24 cannot be meaningfully evaluated under Matbench settings without retraining on MPTrj/sAlex data.
>
> **W2f (Attention optimizations)**
>
> The reviewer is correct that KV-caching and paging are autoregressive techniques and do not apply here. We will remove these from the discussion. The optimizations that do apply to non-autoregressive all-to-all attention are tiled/streaming attention (which we already use via xFormers, capping peak memory below O(N^2)). These are the techniques behind the memory numbers in Fig. 3.
>
> **References**
>
> [1] From Evaluation to Design: Using Potential Energy Surface Smoothness Metrics to Guide Machine Learning Interatomic Potential Architectures. https://arxiv.org/abs/2602.04861
>
> [2] ​​Forces are not Enough: Benchmark and Critical Evaluation for Machine Learning Force Fields with Molecular Simulations https://arxiv.org/abs/2210.07237
>
> [3] The dark side of the forces: assessing non-conservative force models for atomistic machine learning https://arxiv.org/abs/2412.11569

---

> > ### Author Rebuttal · Reviewer_7fTr · 2026-04-02
> >
> > Thank you to the authors for their response. My concerns have been addressed, however I encourage the authors to ensure the remainder of their figures and tables are free of errors, and I reflect Reviewer yfFy's concern regarding non-PBC size extensivity. Nevertheless, I will update my score from 3 to 4.

---

> > > ### Author Response · Authors · 2026-04-04
> > >
> > > We thank the reviewer for re-evaluating and for the updated score. We will audit all figures and tables for correctness in the revision. Regarding non-PBC size extensivity, we agree with Reviewer yfFy and address this in detail in our response.

---

### Official Review · Reviewer_mpXT · 2026-03-13

**Soundness:** 3
**Presentation:** 3
**Significance:** 3
**Originality:** 3
**Overall Recommendation:** 5
**Confidence:** 4

**Summary:**

This paper introduces AllScAIP, an attention-based interatomic potential that combines local neighborhood attention with a global all-to-all node-attention stage. The model also incorporates optional geometric encodings (LAE and ERoPE). The authors argue that many physics-inspired inductive biases—such as rotational equivariance, higher-order angular structure, and radial priors—become increasingly learnable with sufficient model capacity and data scale, while the all-to-all attention stage remains consistently useful for modeling long-range interactions.

Empirically, the paper reports strong results on the OMol25 benchmark, including improvements in energy prediction, long-range distance-scaling tests, and MD-derived observables. Competitive performance is also shown on OMat24 and OC20.

Overall, the architecture is simple and well-motivated, and the scaling-oriented ablation study is interesting because it directly probes how the usefulness of inductive biases changes with data and model size. However, I am not yet fully convinced that the presented evidence supports the paper’s strongest conclusions. While the empirical results are promising, several central claims appear broader than what the current experiments justify.

**Compliance With Llm Reviewing Policy:**

Affirmed.

**Final Justification:**

Authors addressed my concerns. I keep my initial evaluation.

**Key Questions For Authors:**

1) Does the current experimental evidence actually demonstrate that these inductive biases become unnecessary at scale, or only that the specific implementations tested provide diminishing gains in the explored regime?

2) Are the reported scaling trends statistically robust and attributable to scale itself, rather than differences in optimization budget or experimental noise?

3) To what extent are long-range interactions actually learned from data, versus being enabled by the explicitly designed global attention mechanism?

4) Do the current comparisons fairly isolate architectural improvements, or could the observed performance differences be explained by differences in training data, scale, or optimization setup?

**Limitations:**

Yes

**Strengths And Weaknesses:**

Strengths

1. Clear architectural design with a simple and scalable formulation.
The combination of local neighborhood attention with a global all-to-all node-attention stage is conceptually straightforward and well motivated for capturing both local chemical structure and long-range interactions. The design aligns well with recent trends in scaling attention-based architectures.

2. Interesting scaling-oriented ablation study.
A key contribution of the paper is the attempt to analyze how the usefulness of geometric inductive biases changes as both dataset size and model capacity increase. This perspective is valuable, as it directly addresses a broader question in MLIP design: whether physics-inspired biases remain necessary at large scales.

3. Strong empirical performance on OMol25.
The results on OMol25, including energy prediction, distance-scaling behavior, and MD-derived observables, are encouraging. The long-range sensitivity tests are particularly relevant given the paper’s emphasis on global attention mechanisms.

Weaknesses

W1. The main conclusions about “learnability at scale” appear broader than the evidence provided.
The paper repeatedly suggests that rotational equivariance, higher-order directional structure, and even aspects of long-range physics may become largely learnable given sufficient model scale. However, the experiments mainly show that the marginal benefit of the specific encodings tested (LAE and ERoPE) decreases as model and data scale increase, while the global node-attention stage remains beneficial. This supports a claim about the usefulness of these particular encodings in the tested regime, but it does not establish that such inductive biases are generally unnecessary at scale. Notably, the architecture still relies on an explicit all-to-all attention mechanism for global communication, meaning that long-range behavior is not learned implicitly but rather enabled by a dedicated architectural component.

W2. Limited experimental evidence for the scaling claim.
While the observed trend is interesting, the current ablations span a relatively limited set of model and dataset scales. Some performance differences at larger scales appear modest, and the paper does not report variance estimates, multi-seed results, or confidence intervals. As a result, it is difficult to determine whether certain observed reversals are statistically robust. In addition, the comparison between OMol-4M and OMol-102M also changes the training budget, which complicates the interpretation of whether the observed differences arise from scale or optimization budget. More controlled experiments or narrower claims would strengthen the argument.

W3. Evidence regarding rotational equivariance is suggestive but not conclusive.
The paper reports force-rotation consistency after training as evidence that rotational equivariance can emerge from scale. However, this is essentially a numerical sanity check on sampled structures rather than a principled guarantee of equivariance. Moreover, the comparison against random initialization does not provide a particularly informative baseline. As presented, the results indicate that approximate equivariance may emerge in practice, but the broader claim that rotational equivariance becomes learnable at scale may not generalize beyond the tested setting.

W4. The treatment of anisotropic interactions is somewhat unclear.
In the method description, ERoPE is presented as an isotropic, distance-only bias in the global node-attention stage, whereas anisotropic interactions appear to be primarily captured by the local neighborhood-attention stage through directional encodings. If this division of roles is intentional, the paper would benefit from making it more explicit. As written, it is sometimes unclear which aspects of anisotropy are handled locally and which are expected to emerge through global mixing.

W5. Long-range modeling claims may be somewhat overstated.
The experiments provide convincing evidence that all-to-all attention improves long-range behavior on OMol25, including distance-scaling tests and MD observables. However, the paper’s broader claim that explicit long-range physics terms may no longer be necessary is less well supported. Most comparisons are against models such as eSEN, GemNet, and UMA rather than against a broad set of models with explicit long-range components under matched training conditions. Furthermore, the analysis does not fully disentangle improvements due to physically meaningful long-range modeling from more general benefits of graph-wide information propagation or increased capacity on larger systems.

W6. Fairness and clarity of comparisons could be improved.
Several baseline models appear to be trained under different conditions, including different datasets, scales, and conservative/direct-force training regimes. While the paper often specifies training datasets, these differences are not always emphasized in the main comparisons. For example, the OMol25 tables include models trained on OMol-4M, OMol-102M, and UMA-459M datasets, with varying parameter counts and training setups. Similarly, the comparison with UMA involves a model trained on ~500M structures (including additional materials and catalyst data), whereas AllScAIP is trained on ~100M OMol25 data. While the paper acknowledges this difference, the asymmetry makes it difficult to interpret the results as direct state-of-the-art comparisons. A more tightly controlled comparison would strengthen the empirical claims.

---

> ### Author Rebuttal · Authors · 2026-03-31
>
> We thank the reviewer for the detailed feedback.
>
> **W1 + Q1 (Scope of "learnability at scale")**
>
> We agree the claim should be stated precisely. Our evidence shows:
>
> 1. The geometric encodings we tested (LAE, ERoPE) provide diminishing and sometimes negative returns as data and model size grow. These encodings also add computational cost. In our ablations, all variants train for the same number of epochs on the same data, so the no-encoding variant uses less total compute per run. In a compute-limited setting, the simpler model is preferable: it trains faster per step and reaches comparable or better accuracy at the 102M data scale.
>
> 2. The all-to-all node attention is a purely data-driven component with no hand-coded physics, and it remains the most durable source of long-range improvement at every scale we tested. This contrasts with methods like MACELES [1] that inject explicit Ewald-style electrostatics. We compare directly against MACELES on SPICE (App. D) and on 39 molecular liquid MD simulations (Sec. 5.3, Fig. 6): AllScAIP matches or outperforms it on both supervised metrics and experimental observables, without any explicit long-range physics term.
>
> We will tighten the language: our claim is that the tested geometric encodings become less necessary at scale, and that data-driven global mixing can substitute for explicit long-range physics modules in the regimes we evaluated.
>
> **W2 + Q2 (Statistical robustness of scaling trends)**
>
> Running multi-seed experiments at the 102M data scale is prohibitively expensive. Omitting confidence intervals at this scale is standard practice in scaling-law studies. We follow eSEN's training data budget (same data splits, same epochs) for a fair comparison. We will narrow the claim to "in the explored regime" in the revision.
>
> **W3 (Rotational equivariance)**
>
> We agree this test is necessary but not sufficient on its own. Two additional pieces of evidence support the finding:
>
> 1. A concurrent study [2] tests the EScAIP architecture (which AllScAIP builds on) among other non-equivariant models, and shows that this architecture learns 3D-rotational equivariance quickly and reliably, robust to model and dataset size. This is consistent with our observation that AllScAIP achieves 0.9999 force-rotation cosine similarity after training on OMol-102M.
>
> 2. Our MD simulations (on 39 molecular liquids) produce physically correct bulk behavior. Poor rotational equivariance would show up as artifacts in these systems, and we did not observe any.
>
> We will cite [2] and soften the claim to "approximate rotational equivariance emerges in practice, consistent with independent findings on non-equivariant models."
>
> **W4 (Anisotropic interactions)**
>
> The division is intentional. Neighborhood attention operates over directed edges within the local cutoff; when LAE is enabled, it receives explicit angular information. This stage handles anisotropic, short-range interactions (bond angles, dihedral preferences). Node attention operates over all atom pairs with only isotropic distance information (via ERoPE) and handles long-range effects. We will make this division explicit in the revised method section.
>
> **W5 + Q3 (Long-range baselines)**
>
> We compare against MACELES [1], which adds Latent Ewald Summation to the MACE architecture. Both models are trained on SPICE under matched conditions (App. D). AllScAIP achieves lower energy and force MAE across all seven SPICE splits. On 39 molecular liquid MD (Sec. 5.3, Fig. 6), AllScAIP also produces densities and heats of vaporization closer to experiment.
>
> **W6 (Fairness of comparisons)**
>
> We acknowledge the cross-dataset comparisons (OMat24, OC20) involve models trained under different conditions. These results are transfer sanity checks, not leaderboard claims, and we will state this more clearly in the revision. Our controlled experiments are the OMol25 ablations (Table 2), where all variables except the component under test are held fixed.
>
> **Q4 (Isolating architectural improvements)**
>
> On OMol25, the ablations in Table 2 isolate each component (LAE, ERoPE, node attention) at fixed data and model size, so the observed differences are attributable to architecture.
>
> **References**
>
> [1] A Universal Augmentation Framework for Long-Range Electrostatics in MLIPs. https://arxiv.org/abs/2507.14302
>
> [2] Training Dynamics of Learning 3D-Rotational Equivariance. https://openreview.net/forum?id=DLOIAW18W3

---

> > ### Author Rebuttal · Reviewer_mpXT · 2026-04-03
> >
> > The authors addressed my concerns.

---

### Decision · Program_Chairs · 2026-04-30

**Decision:**

Accept (regular)

**Comment:**

Reviewers unanimously recommend accept after a very thorough rebuttal.